# Changing optical properties of Black Carbon and Brown Carbon aerosols during long-range transport from the Indo-Gangetic Plain to the equatorial Indian Ocean

Krishnakant Budhavant[1,4], Mohanan Remani Manoj[2], H.R.C.R. Nair[2], Samuel Mwaniki Gaita[2], Henry Holmstrand[2], Abdus Salam[3], Ahmed Muslim[1], Sreedharan Krishnakumari Satheesh[4], Örjan Gustafsson[2]

[1]Maldives Meteorological Services, 02020, Maldives
[2]Department of Environmental Science and the Bolin Centre for Climate Research, Stockholm University, Stockholm 1069, Sweden
[3]Department of Chemistry, University of Dhaka, Dhaka 1000, Bangladesh
[4]Divecha Centre for Climate Change, Indian Institute of Science, Bangalore 560012, India

*Correspondence to*: Örjan Gustafsson (Orjan.Gustafsson@aces.su.se)

**Abstract.** Atmospheric aerosols strongly influence the global climate by their light absorption (e.g., black carbon, BC, brown carbon, BrC) and scattering (e.g., sulfate) properties. This study presents simultaneous measurements of ambient aerosol light absorption properties and chemical composition from three large-footprint South Asian receptor sites during the South Asian Pollution Experiment (SAPOEX) in December 2017 - March 2018. The BC mass absorption cross-section (BC-MAC$_{678}$) values increased from $3.5 \pm 1.3$ at the Bhola Climate Observatory-Bangladesh

(i.e., located at exit outflow of Indo-Gangetic Plain) to $6.4 \pm 1.3$ at the two regional receptor observatories at Maldives Climate Observatory-Hanimaadhoo (MCOH) and Maldives Climate Observatory-Gan (MCOG), an increase of 80%. This likely reflects a scavenging fractionation resulting in a population of finer BC with higher MAC$_{678}$ having higher longevity. At the same time, the BrC-MAC$_{365}$ decreased by a factor of three from the IGP exit to the equatorial Indian Ocean, likely due to photochemical bleaching of organic chromophores. The high chlorine-to-sodium ratio at the near-

source-region BCOB suggests a significant contribution of chlorine from anthropogenic activities. This particulate Cl$^-$ has the potential to convert into Cl-radicals that can affect the oxidation capacity of the polluted air. Moreover, Cl$^-$ is shown to be near-fully consumed during the long-range transport. The results of this synoptic study over the large South Asian scale contribute rare observational constraints on optical properties of ambient BC (and BrC) aerosols over regional scales away from emission sources. It also contributes significantly to understanding the ageing effect

of the optical and chemical properties of aerosols as the pollution from the Indo-Gangetic Plain disperses over the tropical ocean.

## 1 Introduction

Light-absorbing carbonaceous moieties represent a key component of atmospheric aerosols as they affect global climate due to both their direct absorption and combined/indirect effects with other components (Ramanathan and

Carmichael, 2008; IPCC, 2021). The systematic underestimation of the total optical absorption of aerosols by a factor

of 2 to 3 in climate models compared to observations-based estimates illustrates the current significant uncertainties and potential systematic bias (Gustafsson and Ramanathan, 2016). In addition to climate effects, anthropogenic aerosols such as black carbon (BC) and sulfate ($SO_4^{2-}$) can penetrate deeply into human lungs and increase the risk of cardiovascular and respiratory diseases (Mauderly et al., 2008; Lelieveld et al., 2015; WHO, 2016).

The aerosol loadings in the South Asian region are much higher than the global average, primarily due to anthropogenic activities. The high anthropogenic aerosol levels exert a strong influence on both the climate and the quality of the air people breathe in South Asia, primarily due to massive emissions from the Indo-Gangetic Plain (IGP), the densely populated and industrialized northern part of India and Bangladesh (Shindell et al., 2012; Nair et al., 2023). These anthropogenic aerosols cause a "regional dimming," which reduces the amount of sunlight that reaches the Earth's

surface (Ramanathan et al., 2007; Nair et al., 2023). This, in turn, leads to decreased evaporation and rainfall, which can significantly impact agriculture and water resources (Bollasina et al., 2011). Furthermore, these aerosols have been linked to weakened monsoons responsible for most of the region's rainfall (Jacobson, 2006; Ramanathan et al., 2007). Additionally, the aerosols can intensify tropical storms, making them more destructive (Lin et al., 2023). Perhaps most concerning, anthropogenic carbonaceous aerosols have been linked to the melting of the Himalayan glaciers

(Ramanathan et al., 2007; Ramachandran et al., 2023). This is especially significant because the Himalayas' watershed serves over 3 billion people, making it one of the most important water resources in the world.

    The BC and organic carbon (OC) aerosols are mainly emitted from incomplete fossil fuel combustion and biomass burning (Chakrabarty et al., 2008; Hopner et al., 2016; Dasari et al., 2019). Light-absorbing organic carbon, also known as brown carbon (BrC), consists of water-soluble and water-insoluble components. It is often categorized into water-

soluble and methanol-soluble/water-insoluble BrC to describe its optical properties. BrC is predominantly produced by burning fossil fuels and biomass. It can also be generated via other methods, such as the low-temperature oxidation of biogenic substances, the polymerization of their by-products, reactions involving dienes, and the atmospheric processing of anthropogenic or biogenic volatile organic compounds (VOCs) in the presence of NOx (Andreae and Gelencser, 2006; Laskin et al., 2015).

As per the current understanding, BC displays comparatively low reactivity and undergoes negligible changes over long distances. On the other hand, BrC seems to be subject to bleaching (Dasari et al., 2019). It is, therefore, imperative to delve into the dynamics of the optical properties of BrC during its long-distance transport. Accurate mass absorption cross-section (MAC) and source apportionment of BC aerosols are also crucial as they are input to climate and air quality models (Ram and Sarin, 2015; Gustafsson and Ramanathan, 2016; Venkataraman et al., 2020). BC aerosols

from fossil versus biomass combustion have different light absorption/radiative effects and atmospheric fates (Gustafsson and Ramanathan, 2016; Dasari et al., 2019; Budhavant et al., 2015, 2023). The emissions, source apportionment and optical properties of anthropogenic aerosols from India and the greater South Asia is a key uncertainty in climate and environment research that urgently needs to be addressed.

    Access to the three strategically-located Atmospheric Observatories in South Asia provides an opportunity for synoptic

observations of aerosols along the main wintertime flow trajectory from the key source region for anthropogenic

aerosols to its dispersal over regional scales of the northern Indian Ocean (Figure 1). The arrows in Figure 1 illustrate the common pathway of the well-pronounced South Asian winter monsoon outflow projected from meteorological back trajectory analyses. During the dry winter season with the highest anthropogenic aerosol loadings (e.g., BC, OC, nss-$SO_4^{2-}$, nss-$K^+$), the Himalayas cause topographical steering to force Northern Indian air pollution into the North Bay of Bengal (Figure 1). The main flow is then southward, with many air parcels arriving at Maldives Climate Observatory-Hanimaadhoo (MCOH) and Maldives Climate Observatory-Gan (MCOG).

The South Asian Pollution Experiment 2018 (SAPOEX-18) was a large international campaign to study BC and BrC absorption properties during long-range transport in the South Asian source-receptor system using multiple approaches and sites. The current study reports on the ambient evolution of light-absorption properties for both BC and BrC in connection with the chemical composition of aerosols by combining in situ filter measurements, online optical instrument data of aerosol physical and chemical properties, and satellite and remote sensing data sets. Observations were collected over three strategically located regional receptor sites. The Bhola Climate Observatory-Bangladesh (BCOB) is intercepting the integrated outflow of IGP in rural southern Bangladesh by the shores of the Bay of Bengal. The MCOH in a northern atoll of the Maldives and MCOG situated close to the equator in the southernmost Maldivian atoll are ideal locations for intercepting the larger footprint of the South Asian outflow. Synoptic studies between BCOB and the Indian Ocean receptor sites may shed light on the changing aerosol composition and optical/radiative effects during long-range over-ocean transport. Finally, the observational constraints on the aerosol composition and optical properties are crucial inputs for more accurate modelling of the radiative effects in this large region.

## 2 Methods

### 2.1 Aerosol sample collection

The work presented here was conducted at three sites: BCOB (Lat 22.17ºN Lon 90.71ºE), MCOH (6.78ºN, 73.18ºE), and MCOG (0.69 ºS, 73.15ºE) from early December 2017 to end of March 2021. The BCOB is located on Bhola Island (also called Dakhin Shahbazpur) in the delta of the Bay of Bengal, about 300 km south of Dhaka, Bangladesh (Ahmed et al., 2018; Shohel et al., 2018; Dasari et al., 2019). The MCOH is located in the northern part of Hanimaadhoo island (Thiladhummathi Atoll), around 3.1 km², with around 1800 inhabitants (Corrigan et al., 2006; Hopner et al., 2016; Budhavant et al., 2023). Measurements are taken from a tower platform at 15 m above sea level, from which air samples are directed to a ground-level, air-conditioned laboratory (Corrigan et al., 2006; Budhavant et al., 2018, 2023). The MCOG is located on the southernmost island of the Maldives, at the equator, 500 km south of the capital city Male' and 800 km from MCOH (Corrigan et al., 2006; Ramanathan et al., 2007). A detailed description of each observatory is available in earlier publications (Corrigan et al., 2006; Stone et al., 2007; Dasari et al., 2019). Aerosol $PM_{2.5}$ samples were collected on pre-combusted (at 450 ºC) 150 mm diameter quartz filters (Millipore) using high-volume samplers (DIGITEL Elektronik AG, Model DH77 at 500 liter/minute). Blank filters were shipped, stored, and processed identically as samples. Each of these three observatories are instrumented to record spectral Aerosol Optical Depth (AOD) data under the AErosol RObotic NETwork (AERONET) (Holben et al., 1998; Ramanathan et al., 2005; Nair et al., 2023).

**2.2 Chemical analysis of aerosol filter samples**

The aerosols were analyzed for several carbonaceous components and major ions using standard protocols and suitable techniques (Dasari et al., 2019; Budhavant et al., 2023). The mass concentration of EC (here referred to as BC), OC, and total carbon (TC = BC + OC) were measured with a thermal-optical transmission analyzer (Sunset Laboratory, OCEC analyzer) using the National Institute for Occupational Safety and Health (NIOSH-5040 method) (Birch et al., 1996; Budhavant et al., 2015, 2023). NIST-traceable (Reference Material 8785) laboratory standards verify the accuracy of OC, EC, and TC measurements. No detectable signal was observed for the BC in field blanks. The OC concentration values were blank corrected by subtracting an average field blank (5% of sample signals). Following the established protocol, water-soluble organic carbon (WSOC) was measured using a Shimadzu TOC VCPH analyzer (Kirillova et al., 2010, 2013; Budhavant et al., 2020).

Another portion of each aerosol filter was extracted with 18 M-ohm Milli-Q for analysis of water-soluble inorganic ions by using Ion chromatography (IC, Dionex Aquion, Thermo Scientific). The system contains a guard column and an anion-cation separator column with a primary exchange resign and suppressor column (AERS500/CERS 500). The quality of the data was tested with internal and external reference samples. The analytical error was lower than 4% for the anions and 5% for the cations. A more detailed description can be found in Budhavant et al. (2023).

**2.3 Aerosol Absorption Measurements**

The relationship between atmospheric concentration and direct radiative forcing by BC is its mass absorption cross-section (MAC). The laser beam (678 nm) of the Sunset Laboratory aerosol carbon analyzer was used to measure the light attenuation (ATN = -ln($I/I_0$)) of the aerosols on the filter (Ram and Sarin, 2009). The $MAC_{BC}$ of BC is calculated as (Weingartner et al., 2003; Budhavant et al., 2020)

$$MAC_{BC} = \frac{ATN}{BC_{loading} \cdot MS \cdot R(ATN)} \qquad (1)$$

MS is an empirical multiple scattering correction factor implemented in most filter loading correction schemes. To account for the multiple scattering effects, a factor of 4.5 was selected for estimation (Budhavant et al., 2020). Correction for non-linearity when measuring light absorption through a filter is denoted by R.

$$R = \left(\frac{1}{1.114} - 1\right)\left(\frac{\ln(ATN) - \ln(0.1)}{\ln(0.5) - \ln(0.1)}\right) + 1 \qquad (2)$$

The spectrophotometer measured the light absorption of the aerosol extracted from water. Subsequently, MAC was calculated for WS-BrC.

$$MAC_{WS-BrC} = \frac{b_{abs,365}}{[WSOC]} \qquad (3)$$

where WSOC is the water-soluble organic carbon concentration, $b_{abs,365}$ is the absorption coefficient at 365nm. The absorption Ångström exponent (AAE) was estimated as the slope in a linear regression of the logarithm of the $b_{abs}$ versus the logarithm of the wavelength (λ)

$$\ln|b_{abs}(\lambda)| = -AAE \cdot \ln|\lambda| + \text{intercept} \qquad (4)$$

The AAE was fitted between 330-400 nm to avoid interference from other light-absorbing solutes, such as ammonium
nitrate, sodium nitrate, and nitrate ions (Cheng et al., 2011; Bosch et al., 2014).

**2.4 Aerosol Radiative Forcing**

The radiative forcing of aerosol particles is a major uncertainty factor in understanding the Earth's climate (Ramanathan et al., 2007; IPCC et al., 2021; Lu et al., 2023). The radiative implications of aerosols are quantified in
terms of their Direct Aerosol Radiative Effects (DARE). Spectral Aerosol Optical Depth (AOD) data from three stations under the AErosol RObotic NETwork (AERONET) (Hess et al., 1998; Bedareva et al., 2014), ozone (OMI, Ozone Monitoring Instrument), water vapor and surface reflectance (MODIS, Moderate Resolution Imaging Spectroradiometer) and surface reflectance were used in this study. The aerosol optical model (Optical Properties of Aerosols and Clouds, OPAC 3.1), which works based on Mie scattering theory (Hess et al., 1998), was used to estimate
the optical properties of newly defined aerosol mixtures (Hess et al., 1998). The AOD from the sun photometers, single scattering albedo (SSA), and asymmetry parameters modeled using the Mie scattering model were used as input to Santa Barbara DISORT Atmospheric Radiative Transfer (SBDART) (Ricchiazzi et al., 1998; Lu et al., 2023). The model uses the complex discrete ordinate method to numerically integrate the radiative transfer equations (Stamnes et al., 1988). A detailed description of this model and approach is available elsewhere (Ramanathan et al., 2005; Satheesh
et al., 2002; Nair et al., 2023).

**2.5 Air mass back-trajectories and remote sensing**

This study examines air mass back trajectories to identify potential sources of BC and other aerosol components arriving at the MCOG station, which is even further away than MCOH from source regions. Given this long-distance
travel, the focus on the dry season (increasing longevity), that BC has longer lifetimes than other aerosols, and the experiences from earlier studies (Budhavant et al., 2020, 2023), a BT time horizon of ten days was selected as appropriate. The AMBTs were generated at an arrival height of 50m at all three sampling sites (Supplementary (S) figures S1–S4), the NOAA Hybrid Single-Particle Lagrangian Trajectory model (version 4) (Draxler et al., 1997; Draxler, 1999). This study's calculations were based on meteorological data from the Global Data Assimilation System
(GDAS). The GDAS is run four times daily (at 0000, 0600, 1200, and 1800 UTC). These individual trajectories were clustered into different geographical regions (Figure 1). The MODIS (Moderate Resolution Imaging Spectroradiometer) satellite-derived FIRMS (Fire Information for Resource Management System) based fire-count

data combined with cluster analysis to understand the impact of biomass burning emissions from potential source regions during the sampling period (Figure 1).

## 3 Results and Discussion

### 3.1 Atmospheric Transport

The AMBTs, AOD, and active fire data were used as parameters to study the atmospheric transport and geographical source regions in the area. Based on atmospheric transport, we defined two temporal source domains: The influence of the heavily polluted Indo Gangetic Plain (IGP) region (18 December 2017 to 8 February 2018) and the total period of the study. Measurements at BCOB represent an accumulation of IGP sources through air mass transport across N. Pakistan, N. India, and Bangladesh, a region containing many large- and mega-cities, regions of heavy industrialization and rural areas with extensive agricultural burning (Figure 1 and Figure S2). The MCOH and MCOG are situated in the northern Indian Ocean and thus intercept long-range pollutant emissions from South Asia, including the IGP, the western part of India, and the Indian Ocean (Figure S3 and Figure S4). Occasionally, the winds in the IGP sector come from southern India or the Bay of Bengal. However, during winter, polluted winds from the IGP can reach the Bay of Bengal, leading to similar signals being detected over the MCOH and MCOG regions. This is particularly noticeable during synoptic observations. Cluster analysis of AMBTs combined with AOD, satellite measurements, and aerosol chemical composition demonstrated that the wintertime northern Indian Ocean is greatly influenced by anthropogenic aerosols transported from source regions like IGP and the western margin of India.

### 3.2 Organic and Black Carbon

In general, varying primary and secondary sources combined with short atmospheric residence times of aerosol particles containing a high fraction of organic carbon result in large regional differences in chemical composition, morphology, mixing state, size, and optical properties. OC was the main component of the carbonaceous aerosol in South Asian winter, accounting for $85 \pm 5\%$ of total carbon (TC) at BCOB, $66 \pm 9\%$ at MCOH, and $67 \pm 9\%$ at MCOG. The OC contribution to TC was highest when the wind came from IGP (Figure 2) and minimum when the wind traveled through oceanic regions at all three sites. The BC and OC are very well correlated (R > 0.74, Table S1-S3) at all three sampling sites, indicating similar source emissions. However, the average ratio of OC to BC was $6.5 \pm 2.1$ at BCOB, decreasing markedly to $2.2 \pm 1.1$ at MCOH and $2.4 \pm 1.8$ at MCOG. This large decrease from the exit of the IGP source region (BCOB) to the Indian Ocean receptor sites (MCOH and MCOG) demonstrates that OC/BC ratios were strongly affected by selective processing and/or washout of OC during long-range transport (LRT). The atmospheric lifetime of OC is typically shorter than that of BC in this region (Budhavant et al., 2020). Since OC also represents a more complex mixture, it is subject to more atmospheric transformation than BC, reflected in a larger shift in stable isotope fingerprints of the OC component from source to receptor sites in this region (Dasari et al., 2019; Bosch et al., 2014; Kirillova et al., 2016). The highest concentrations of BC, OC, and nss-$SO_4^{2-}$ aerosols were associated with air masses from IGP and the western margin of India (Tables 1-2, and S4).

The water-insoluble fraction of BrC exhibits a higher absorption rate per unit of mass than the WS fraction (Liu et al., 2013; Cheng et al., 2016). We observed that WSOC comprised a large but declining proportion of the overall OC (Figure 3). The WSOC/OC ratio changed throughout the study, as shown in Figure 4C. At BCOB, the WSOC/OC ratio (0.35 ± 0.06) was less variable, indicating that the sources of both types of carbon were similar. Furthermore, we discovered a strong correlation between WSOC and OC concentrations at BCOB (r= 0.95, p<0.001). However, the lower WSOC/OC ratios at MCOH (0.21 ± 0.1) and BCOB (0.16 ± 0.1) suggest a higher contribution of water-insoluble OC at these locations and time. Carbonaceous aerosols derived from fossil fuel combustion may be relatively less water-soluble (WSOC ≥20%) due to less oxygenated organic moieties (Ruellan et al., 2001). The mass fraction of WSOC to OC was observed as an indicator of aerosol photochemical processing in the atmosphere (Dasari et al., 2019). Taken together, the decreasing OC/BC between IGP exit and after transportation over the ocean, indicate selective washout and bleaching reactions of organic carbon.

### 3.3 Characteristics of the ionic aerosol components

The chemical composition of the aerosols changed both between sites and over time (Table 1, Figure 2). Filter samples were characterized in terms of major anions ($Cl^-$, $NO_3^-$, and $SO_4^{2-}$) and major cations ($Na^+$, $K^+$, $Mg^{2+}$, $Ca^{2+}$, and $NH_4^+$) for the four-month sampling period (Figure 4). The highest concentrations of ions were noted for BCOB, which is expected as the site is situated at the outflow of highly polluted IGP. However, $SO_4^{2-}$ and $NH_4^+$ followed another pattern. Higher values for $SO_4^{2-}$ and $NH_4^+$ were found at MCOH. These MCOH-intercepted pollutants were traced to India's central and eastern regions and the IGP through AMBTs. The IGP region and surrounding areas are hotspots for sulfur dioxide ($SO_2$) emissions, mainly due to multiple thermal power plants, construction industries, and petroleum refineries. These sources contribute to the region's $SO_2$ and nitrogen oxides (NOx) (Guttikunda et al., 2014; Kuttippurath et al., 2022). Furthermore, a previous study at MCOH found that dimethyl sulfide (DMS) contributes only up to 3% to nss-$SO_4^{2-}$ in polluted air (Granat et al., 2010). The IGP is a hotspot of high anthropogenic aerosol loading due to intense agriculture crop residue burning, biomass burning, open waste burning, industries, and high urban activities (Dasari et al., 2020; Ansari and Ramachandran, 2023; Nair et al., 2024).

To identify the effect of marine influences on aerosol composition, sea salt corrections were calculated using $Na^+$ as the reference element (Keene et al., 1986). The nss-$SO_4^{2-}$ fraction to total sulfate was at BCOB (99 ± 1 %, mean ± standard deviation), MCOH (98 ± 1%), and MCOG (86 ± 13%), indicating significant contributions of $SO_4^{2-}$ and $SO_2$ from diesel combustion and coal-fired power plants in India and Bangladesh. Some of the nss-$SO_4^{2-}$ at MCOH may be due to ocean traffic over the northern Indian Ocean, as the majority of shipping emissions result from the combustion of its fuel that releases $SO_x$ (Sulphur Oxides) and $NO_x$ directly into the atmosphere (Corbett and Koehler, 2003; Gopikrishnan and Kuttippurath, 2021).

The near-IGP-source-region BCOB was discovered to have a high $Cl^-/Na^+$ ratio (Figure 4, 4.7 ± 3.5) compared to the other two receptor sites (figure 4). This implies that a significant amount of total $Cl^-$ comes from anthropogenic activities at BCOB. The particulate $Cl^-$ might come from the burning of plastics containing chlorine, such as polyvinyl

chloride (PVC), in open waste burning (Pathak et al., 2023), which can be converted into $Cl^-$ radicals that impact the oxidation capacity of the polluted air. Marine aerosol often experiences chlorine depletion, and releasing gas-phase HCl from particles can impact aerosol acidity and the concentration of water-soluble ions. However, once these particles enter the atmosphere, they become exposed to various pollutants, leading to the loss of particulate chlorine into the gaseous phase. This loss of chlorine is typically attributed to ion exchange reactions with atmospheric acids like $SO_2$, $H_2SO_4$, and $HNO_3$, which result in the formation of sulfates and nitrates, as well as the degassing of HCl (Orsini et al., 1986; Brimblecombe and Clegg, 1988; Haslett et al., 2023). Other pathways lead to the loss of particulate chlorine, such as interaction with NO: $N_2O_5$, HOBr, and $O_3$, as well as the release of NOCl, HONO, $ClNO_2$, $Cl_2$, and BrC (Vogt et al., 1996; Behnke and Zetzsch, 1989; Haslett et al., 2023). These pathways can have significant implications for marine tropospheric chemistry and the polluted coastal atmosphere due to the creation of photochemically active halogenated gaseous compounds. The study found significant anthropogenic chloride emissions from human activities, which can affect the oxidation capacity of polluted air.

We observed a high correlation ($r \geq 0.7$) between BC, OC, and nss-$K^+$ in aerosol samples collected at BCOB (Table S1) and MCOH (Table S2). The nss-K fraction to total potassium was observed at BCOB ($97 \pm 3$ %), MCOH ($78 \pm 13\%$), and MCOG ($42 \pm 33\%$), indicating significant contributions from biomass burning at BCOB and MCOH, as nss-$K^+$ is considered as a proxy for identifying the regional impact of biomass burning emissions (Andreae, 1983; Paris et al., 2010). High concentrations of nss-$SO_4^{2-}$, nss-$K^+$, and $NH_4^+$ in measured ions and carbon aerosols indicate strong anthropogenic sources in the ambient aerosols over the northern Indian Ocean.

Our observations have shown an increase in the $SO_4^{2-}$/BC ratio when aerosols are transported from IGP. Notably, this ratio is more pronounced at MCOH than at BCOB (Figure 5). This shift in composition likely signifies the generation of secondary sulfate from anthropogenic $SO_2$ during extended transportation. It was observed that there was a lower $SO_4^{2-}$/BC at MCOG than at MCOH. This might be because $SO_4^{2-}$ gets washed out more easily than BC in this region (Budhavant et al., 2020), and another factor is that MCOG has slightly different AMBT paths than MCOH (Figure 1). It is worth noting that there could be some minor sources of emissions along the route to the receptor sites, like ships and small islands. However, their impact on the overall regional loading of BC is insignificant. Therefore, we can infer that most BC loading over the northern Indian Ocean originates from high-emission source areas in South Asia. While there was an increase in the $SO_4^{2-}$/BC ratio, two other important coating components, WSOC/BC and WIOC/BC, declined.

### 3.4 Black carbon mass absorption cross-section

The impact of BC aerosols on air quality, boundary layer dynamics, and climate depends not only on BC concentration but also on the light absorption characteristics of BC. Moreover, MAC values are crucial to estimate radiative forcing accurately. The MAC of BC is here denoted as "BC-$MAC_{678}$" During the SAPOEX-18 campaign, the calculated BC-$MAC_{678}$ was found to have a lower average value at BCOB ($4.4 \pm 1.9$ $m^2g^{-1}$) and higher at the most distant Indian Ocean receptor station MCOG ($7.0 \pm 1.9$ $m^2g^{-1}$), with MCOH, at shorter over-ocean transport distance having a value

of $6.1 \pm 1.3$ m$^2$g$^{-1}$ (Figure 2). A recent laboratory-based study of the coating-enhancement of BC-MAC (E-MAC) from these observatories showed that the E-MAC was about 1.6 (i.e., a 60% enhancement in net BC-MAC from coating effects) at all three stations. This constancy suggests that the coating-aging effect of BC was near-complete already upon arrival at BCOB (Nair et al., 2024). The approximately 80% increase in BC-MAC$_{678}$ during over-ocean transport therefore must signal another mechanism. It likely reflects a selective fractionation of the BC population whereby larger and less hydrophobic BC is preferentially scavenged, whereas a finer pool with higher BC-MAC$_{678}$ is becoming relatively more prevalent by arrival to the distant Indian Ocean receptor observatories. This is consistent with an earlier finding during the winter at MCOH, where it was observed that BC-MAC$_{678}$ in the rain was lower than BC-MAC$_{678}$ measured for suspended aerosols collected simultaneously from the air (Budhavant et al., 2020). By shedding light on the aging effect of the optical properties of BC aerosols, the study results advance our understanding of this important topic.

These observational constraints in the South Asian global hotspot region are consistent with global simulation models that suggest that in ~1-5 days, the BC can internally mix with other aerosols (Jacobson et al., 2000). After mixing, the photochemical properties of pure BC will no longer be retained due to the coating of the other aerosols in the atmosphere, such as sulfate, nitrate, and organics. Observational data on BC-MAC in an ambient atmosphere far from immediate sources is rare. Most models use laboratory-based or city measurements for MAC and E-MAC values. (e.g., Bond and Bergstrom, 2006; Wang et al., 2016). This study can help bridge the gap between model underestimations and observational estimates of BC and AAOD pointed out for South Asia (Gustafsson and Ramanathan, 2016). These findings can be utilized to refine model estimates of radiative forcing from both BC and BrC for the large-emission region of South Asia. The severe air pollution from the IGP spreads over large regional scales over the Indian Ocean. Therefore, understanding the ageing effect of the optical and chemical properties of aerosols is significant, particularly for this region.

**3.5 Light Absorption Properties of Brown Carbon**

In addition to BC, BrC also affects the radiative forcing at ultraviolet wavelengths, although its MAC is an order of magnitude less than BC in the visible wavelength range (Bosch et al., 2014; Kirillova et al., 2013). During the campaign, measurements of WS extracts of BrC show significant differences in light absorption characteristics between the three sampling sites. The average MAC measured at 365 nm (BrC MAC$_{365}$) at BCOB ($1.0 \pm 0.3$ m$^2$ g$^{-1}$) was two to three times higher than that measured at MCOH ($0.3 \pm 0.3$ m$^2$ g$^{-1}$) and MCOG ($0.6 \pm 0.3$ m$^2$ g$^{-1}$) (Figure 4). BrC MAC$_{365}$ measured during this study is broadly in the same range as earlier studies focusing on fewer locations from the same region (Bikkina et al., 2014; Dasari et al., 2019). Primary BrC emitted from biomass burning appears to be more light absorptive than secondary aerosols; MAC values at 405 nm ranged from 0.2 to 1.5 m$^2$ g$^{-1}$ for humic and fulvic acids and 0.001 to 0.09 for secondary organic aerosols (Lambe et al., 2013), while Chen and Bond (2010) reported a range for primary aerosols from 0.1 to 1.1 m$^2$ g$^{-1}$ (Chen et al., 2010). The average BrC MAC$_{365}$ measured during this study was lower than values reported from close to sources in megacities such as the 1.8 m$^2$ g$^{-1}$ for Beijing

winter (Cheng et al., 2011), $1.6 \pm 0.5$ m$^2$ g$^{-1}$ for Delhi (Kirillova et al., 2014), $1.6 \pm 0.1$ m$^2$ g$^{-1}$ for Kanpur (Choudhary et al., 2016), $1.5 \pm 0.2$ m$^2$ g$^{-1}$ for Kathmandu (Chen et al., 2020). This indicates that the MAC-BrC decreased by a factor of three from the IGP exit to the equatorial Indian Ocean (Figure 3).

The AAE characterizes the spectral characteristic of BrC. Furthermore, AAE is often used to characterize BrC from coal combustion, biomass, and biofuel burning (Chen and Bond, 2010; Rastogi et al., 2021). The AAE value of BrC is typically reported to be ~1 (fossil fuel emissions), ~7 (biomass burning), and 7-15 for laboratory-generated smoke and smoldering of different types of woods (Hoffer et al., 2006; Chen et al., 2010). The average values of AAE (330-400 nm range) of WS-BrC intercepted at the South Asian receptor observatories were $5.5 \pm 2.7$ at BCOB, $6.5 \pm 2.4$ at

MCOH, and $4.1 \pm 0.5$ at MCOG. These compare to AAE values measured at Nepal Climate Observatory-Pyramid ($4.9 \pm 0.7$, Kirillova et al., 2016), in New Delhi in winter ($5.1 \pm 2.0$, Kirillova et al., 2014). However, the AAE values in this study show a difference in values but not significant as previously measured at MCOH in winter ($7.2 \pm 0.7$, Bosch et al., 2014) and in the IGP outflow measured over the Bay of Bengal ($9.1 \pm 2.5$, Bikkina and Sarin, 2013). The results of this synoptic study at a large scale in South Asia are significant for understanding the aging effect of optical and

chemical properties of aerosols.

**3.6 Aerosol Radiative Forcing**

From December to March, the tropical Indian Ocean/atmosphere system provides a natural opportunity to study aerosol radiative forcing influenced by anthropogenic aerosols (Satheesh and Ramanathan, 2000; Nair et al., 2023). This is

due to the fact that the Indian Ocean atmosphere receives polluted air that travels from the Indian subcontinent and surrounding regions (Figure 1, Gustafsson et al., 2009; Budhavant et al., 2018).

The DARF (cloud-free atmosphere) has been estimated over BCOB, MCOH, and MCOG on a monthly basis using aerosol optical properties obtained from the OPAC model in SBDART (Figure 5). The DARF at the top of the atmosphere (TOA, 3 km) and at the surface is calculated by estimating the difference of downward and upward fluxes

simulated by the model in the atmosphere conditions with and without aerosols for the three sites. The surface forcing and TOA were both negative at all three sampling stations, indicating cooling effects. Meanwhile, the atmospheric column represented a warming effect. The negative sign depicts the dominant presence of scattering aerosols.

The study observed that the average atmospheric forcing at BCOB was higher ($10.5 \pm 3.2$ Wm$^{-2}$) compared to MCOG ($4.8 \pm 2.1$ Wm$^{-2}$) and MCOH ($8.0 \pm 2.6$ Wm$^{-2}$). This difference was due to anthropogenic aerosols (Table S4),

particularly BC (net absorbing), which caused a warming effect in the atmosphere (Figure 2 and Figure 5). BCOB experienced nearly double atmospheric forcing compared to the remote equatorial ocean at MCOG, likely attributed to its higher concentration of anthropogenic aerosol loading, such as BC, NO$_3^-$, and nss-K$^+$ (Table S4, Figure 3). During winter, the IGP experiences a significant increase in aerosol loading, mainly carbon aerosols resulting from fossil fuel and biofuel combustion (Gustafsson et al., 2009; Kaskaoutis et al., 2014; Dasari et al., 2020). As spring progresses,

dust becomes the dominant aerosol in the northwestern region of India and arid areas of southwest Asia (Kaskaoutis et al., 2014; Singh et al., 2014; Dumka et al., 2023). At the same time, significant agricultural burning in Southeast

Asia results in significantly elevated concentrations of carbonaceous aerosols (Kaskaoutis et al., 2014; Budhavant et al., 2015; Bikina et al., 2019). Also, Himalayan forest fires and wheat residue burning in the IGP contribute to the aerosol burden during spring (Gautam et al., 2007; Bikina et al., 2019). BCOB experienced a high atmospheric forcing in March, particularly in the outflow region of the IGP (Figure 5). In January, MCOH experienced slightly higher atmospheric forcing (11.2 $Wm^{-2}$) than BCOB (10.4 $Wm^{-2}$). The findings are consistent with our earlier study conducted at MCOH, which showed that anthropogenic aerosols, such as BC, OC, nss-$K^+$, nss-$SO_4^{2-}$, and $NH_4^+$, were predominantly in the fine mode (70-95%) and particularly observed in the air masses coming from IGP during the period (Budhavant et al., 2018). Therefore, it is crucial to address this issue and take appropriate measures to reduce the amount of anthropogenic aerosol loading.

## 4 Summary

The South Asian Pollution Experiment 2018 (SAPOEX-18) utilized access to three strategically located atmospheric receptor observatories to provide synoptic observations of the optical properties of ambient carbonaceous aerosols along the main wintertime flow trajectory from key source regions. The increase in BC-$MAC_{678}$ from the IGP outflow of BCOB to the receptor stations (MCOH and MCOG) was about 80%. As earlier reports for this system have demonstrated that there is no additional enhancement in BC-MAC from aerosol coatings during LRT from BCOB, this likely reflects a scavenging fractionation resulting in a population of finer BC with higher $MAC_{678}$ having higher lifespan. These observational constraints revealed opposite trends during long-range transport in BC-MAC and BrC-MAC (decreasing, presumably due to photochemical bleaching). The study also found significant anthropogenic chloride emissions from human activities, which can affect the oxidation capacity of polluted air. Models estimating the climate effects of particularly BC aerosols may have underestimated the ambient BC-MAC over distant and extensive receptor areas, which could contribute to the discrepancy between aerosol absorption predicted by models constrained by observations. These findings can be utilized to refine model estimates of radiative forcing from both BC and BrC for the large-emission region of South Asia. This is particularly relevant as the severe air pollution from the Indo-Gangetic Plain spreads over large regional scales over the Indian Ocean.

## Competing interests

The authors declare that they have no conflict of interest.

## Author Contributions

K.B. and Ö.G. conceived the study. K.B. collected the Samples at MCOH, A.S. was responsible for sample collection at BCOB, and A.M. was responsible for sample collection at MCOG. K.B. performed the chemical analysis, with the support of S.G., E.K., H.N., and M.R., performed the radiative forcing estimations and analyzed satellite data. K.B. and Ö.G. interpreted the data and drafted the manuscript. All co-authors provided input on interpretations and early versions of the manuscript.

## 5. Acknowledgment

Elena Kirillova and Sanjeev Dasari (Stockholm University) are acknowledged for their support during the field campaign. We thank the technical staff at BCOB, MCOH, and MCOG for their continued field support. A special thanks to the Maldives Meteorological Services and the Government of the Republic of Maldives for the ongoing support of the joint MCOH-MCOG operation. K.B. thanks to the additional support from the Regional Resources Center for Asia and the Pacific (RRC.AP), Asian Institute of Technology (AIT), Thailand. We acknowledge financial

support from the Swedish Research Council for Sustainable Development (FORMAS Contract Nr. 2020-01917) and the Swedish Research Council (VR Contract Nos 2017-01601, and 2020-05384).

**Table 1.** Aerosol optical depth (AOD), mass absorption cross-section (MAC) of black carbon (BC) and brown carbon (Br-C), the concentration of BC, organic carbon (OC), water-soluble organic carbon (WSOC), and absorption Angstrom exponent (AAE) were measured at the Bhola Climate Observatory-Bangladesh (BCOB), Maldives Climate Observatory- Hanimaadhoo (MCOH), Maldives Climate Observatory-Gan (MCOG) during November 2017 to March 2018.

| Site | AOD | BC-MAC$_{678}$ (m$^2$ g$^{-1}$) | BrC-MAC$_{365}$ (m$^2$ g$^{-1}$) | BC (μg m$^{-3}$) | OC (μg m$^{-3}$) | WSOC (μg m$^{-3}$) | AAE$_{BrC}$ (330-400nm) |
|------|-----|------|------|------|------|------|------|
| **BCOB** | $0.8 \pm 0.3$ | $4.4 \pm 1.9$ | $1.0 \pm 0.3$ | $3.0 \pm 1.3$ | $20 \pm 11$ | $6.9 \pm 4.0$ | $5.5 \pm 2.7$ |
| **MCOH** | $0.5 \pm 0.2$ | $6.1 \pm 1.3$ | $0.3 \pm 0.3$ | $1.0 \pm 0.5$ | $2.3 \pm 1.5$ | $0.5 \pm 0.4$ | $6.5 \pm 2.4$ |
| **MCOG** | $0.2 \pm 0.1$ | $7.0 \pm 1.9$ | $0.6 \pm 0.3$ | $0.3 \pm 0.3$ | $0.7 \pm 0.5$ | $0.1 \pm 0.1$ | $4.1 \pm 0.5$ |
| Only during synoptic period (18 December 2017 to 8 February 2018) | | | | | | | |
| **BCOB** | $0.9 \pm 0.4$ | $3.5 \pm 1.3$ | $1.0 \pm 0.2$ | $3.6 \pm 1.0$ | $27 \pm 8.7$ | $9.1 \pm 2.8$ | $6.4 \pm 2.0$ |
| **MCOH** | $0.5 \pm 0.2$ | $6.4 \pm 1.3$ | $0.2 \pm 2.0$ | $1.1 \pm 0.5$ | $3.0 \pm 1.6$ | $0.6 \pm 0.4$ | $7.6 \pm 1.5$ |
| **MCOG** | $0.3 \pm 0.2$ | $6.4 \pm 1.7$ | $0.7 \pm 0.2$ | $0.5 \pm 0.2$ | $0.9 \pm 0.4$ | $0.1 \pm 0.1$ | $4.0 \pm 0.9$ |

**Table 2.** Concentrations of major ions (µg m$^{-3}$) were measured at the Bhola Climate Observatory-Bangladesh (BCOB), Maldives Climate Observatory-Hanimaadhoo (MCOH), Maldives Climate Observatory-Gan (MCOG) from November 2017 to March 2018.

| Site | Na$^+$ | Cl$^-$ | NO$_3^-$ | NH$_4^+$ | nss-SO$_4^{2-}$ | nss-K$^+$ | Ca$^{2+}$ |
|------|--------|--------|----------|----------|-----------------|-----------|-----------|
| **BCOB** | 0.4 ± 0.3 | 1.9 ± 1.8 | 7.6 ± 7.3 | 3.8 ± 3.2 | 11 ± 5 | 2.4 ±1.2 | 0.1 ± 0.1 |
| **MCOH** | 0.8 ± 0.5 | 0.7 ± 0.5 | 0.1 ± 0.0 | 4.2 ± 2.7 | 11 ± 7 | 0.4 ± 0.3 | 0.1 ± 0.0 |
| **MCOG** | 1.0 ± 0.9 | 0.8 ± 0.8 | 0.3 ± 0.4 | 0.5 ± 0.8 | 3.0 ± 2 | 0.1 ± 0.1 | 0.2 ± 0.2 |
| Only during the synoptic period (18 December 2017 to 8 February 2018) | | | | | | | |
| **BCOB** | 0.5 ± 0.4 | 3.0 ± 1.9 | 12 ± 7.7 | 2.5 ± 3.5 | 12 ± 5 | 2.9 ± 1.0 | 0.1 ± 0.1 |
| **MCOH** | 1.0 ± 0.5 | 0.6 ± 0.4 | 0.1 ± 0.0 | 4.8 ± 3.6 | 16 ± 7 | 0.5 ± 0.3 | 0.1 ± 0.0 |
| **MCOG** | 1.1 ± 1.1 | 0.6 ± 0.4 | 0.3 ± 0.3 | 0.9 ± 0.1 | 4.7 ± 4 | 0.1 ± 0.1 | 0.2 ± 0.2 |

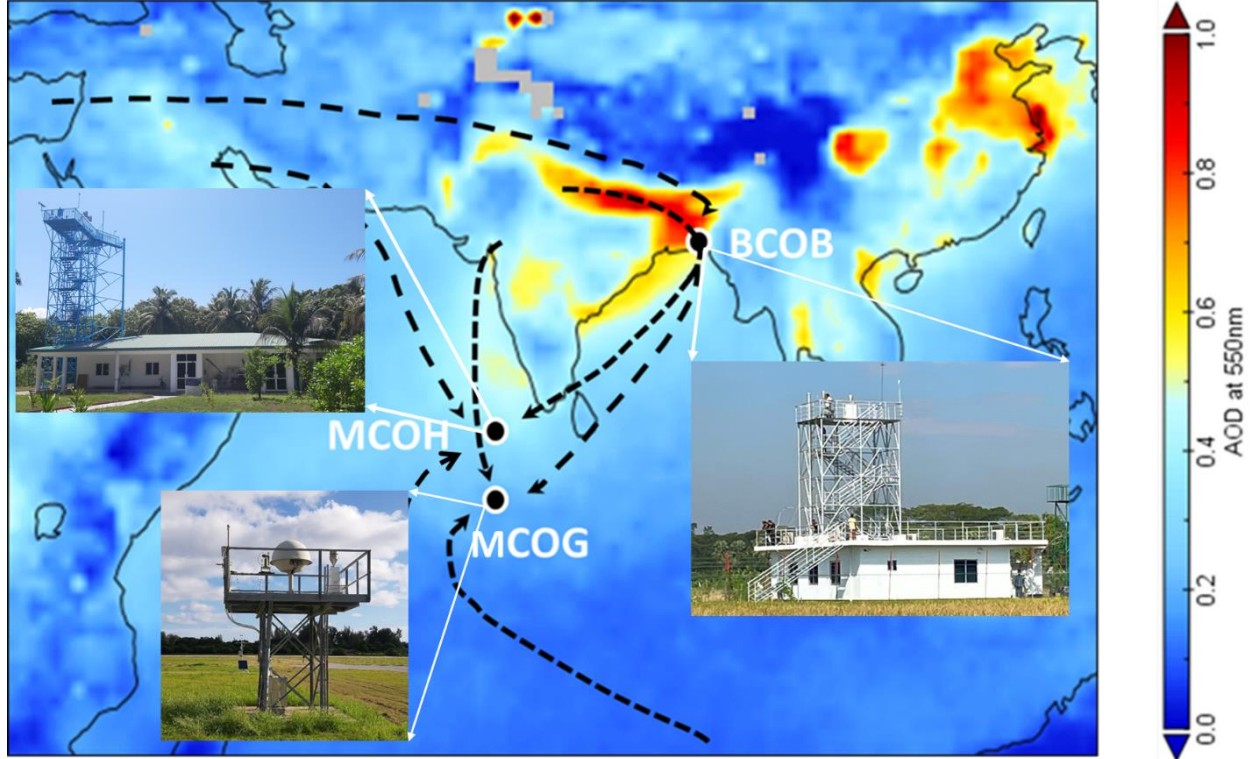

**Figure 1.** Average Aerosol Optical Depth (AOD) at 550 nm from Moderate Resolution Imaging Spectroradiometer (MODIS) during SAPOEX-18 from December 2017 to March 2018 over the South Asian region. The receptor sites are shown (black fill, with pictures): the Bhola Climate Observatory in Bangladesh (BCOB), the Maldives Climate Observatory at Hanimaadhoo (MCOH), and the Maldives Climate Observatory at Gan (MCOG). The thick black lines with arrow show mean air mass trajectory clusters (more details in Supporting Information Figures S1 and S4).

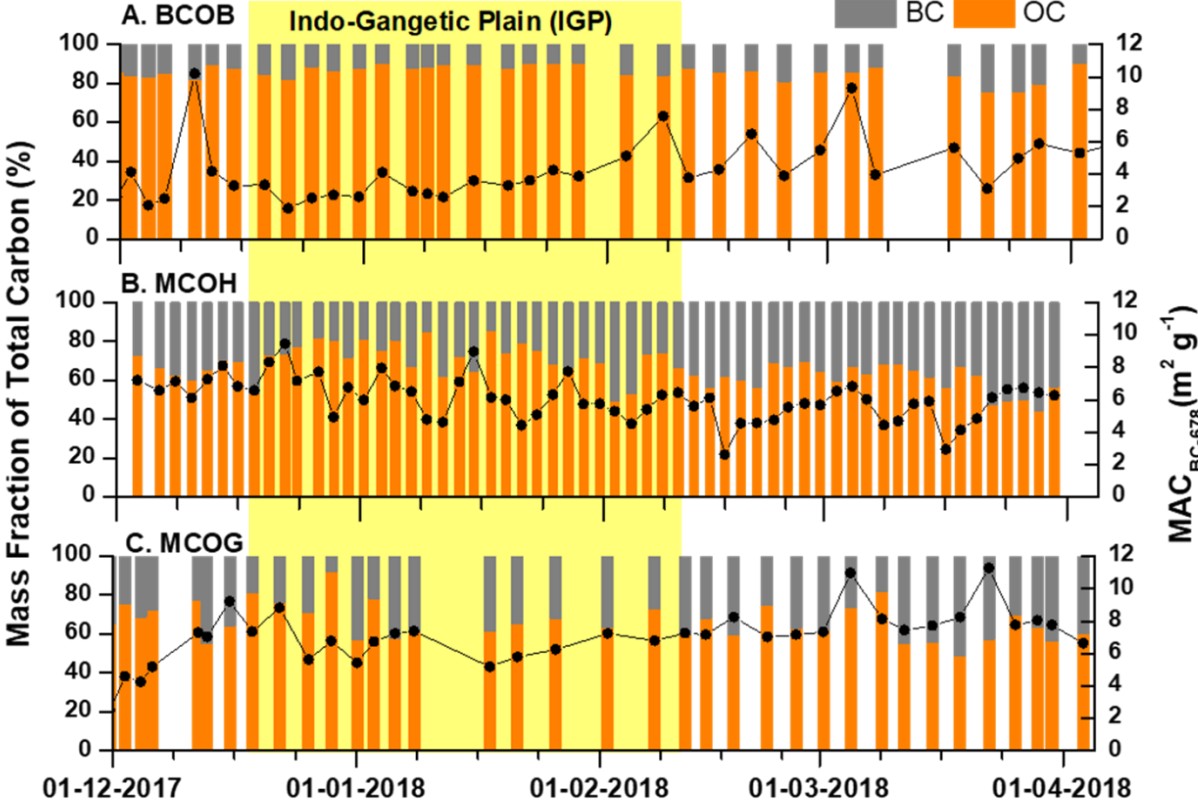

**Figure 2.** Mass fraction of total carbon (black carbon + organic carbon) and BC mass absorption cross-section (BC-MAC, at 678nm) was measured at three receptor sites in South Asia, i.e., A. Bhola Climate Observatory-Bangladesh (BCOB), B. Maldives Climate Observatory-Hanimaadhoo (MCOH) and C. Maldives Climate Observatory-Gan (MCOG) from 1 December 2017 to early April 2018. The vertical yellow field indicates predominance of air mass origin from the high-pollution source region Indo-Gangetic Plain (IGP).

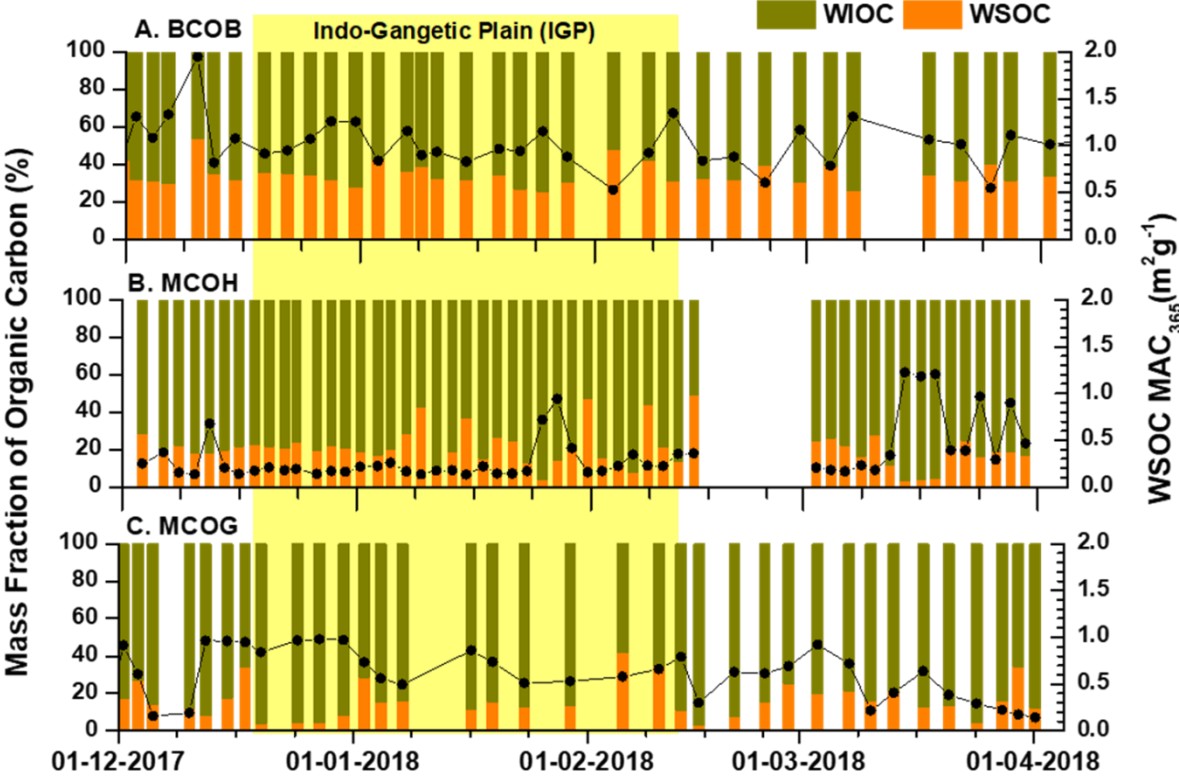

**Figure 3.** Mass fraction of organic carbon (divided as water-insoluble organic carbon vs water-soluble organic carbon) and mass absorption cross-section for Brown Carbon (Br-C MAC, at 365 nm) measured at three receptor sites in South Asia, i.e., A. Bhola Climate Observatory-Bangladesh (BCOB), B. Maldives Climate Observatory-Hanimaadhoo (MCOH) and C. Maldives Climate Observatory-Gan (MCOG) from 1 December 2017 to early April 2018. The vertical yellow bar indicates predominance of air mass origin from the high-pollution source region Indo-Gangetic Plain (IGP).

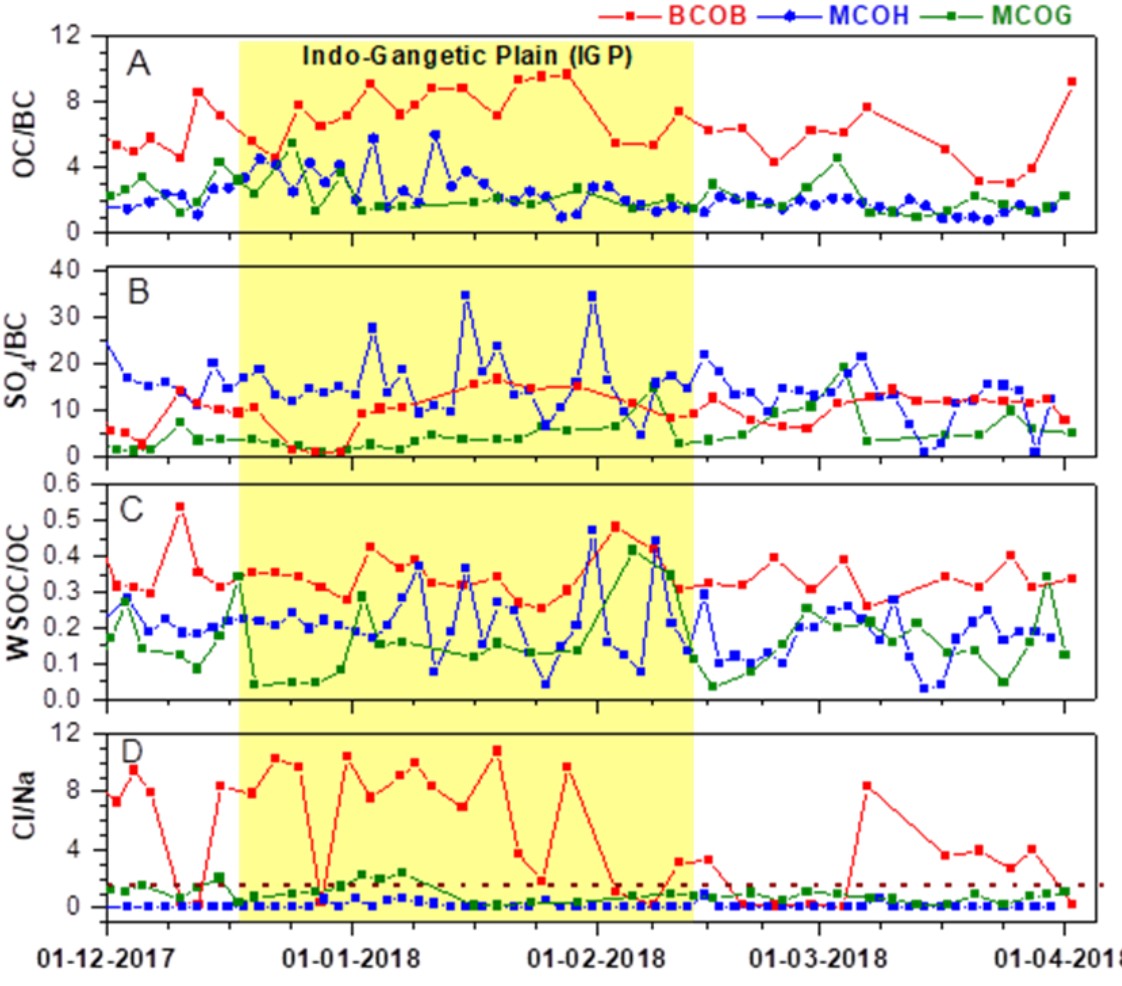

420

**Figure 4.** Time series of the ratio of measured chemical species OC/EC (panel A), SO$_4$/BC (panel B), WSOC/BC (panel C), and Cl/Na (panel D, seawater ratio 1.8, dotted line) over three receptor sites in South Asia: Bhola Climate Observatory-Bangladesh (BCOB), Maldives Climate Observatory-Hanimaadhoo (MCOH), and Maldives Climate Observatory-Gan (MCOG).

425

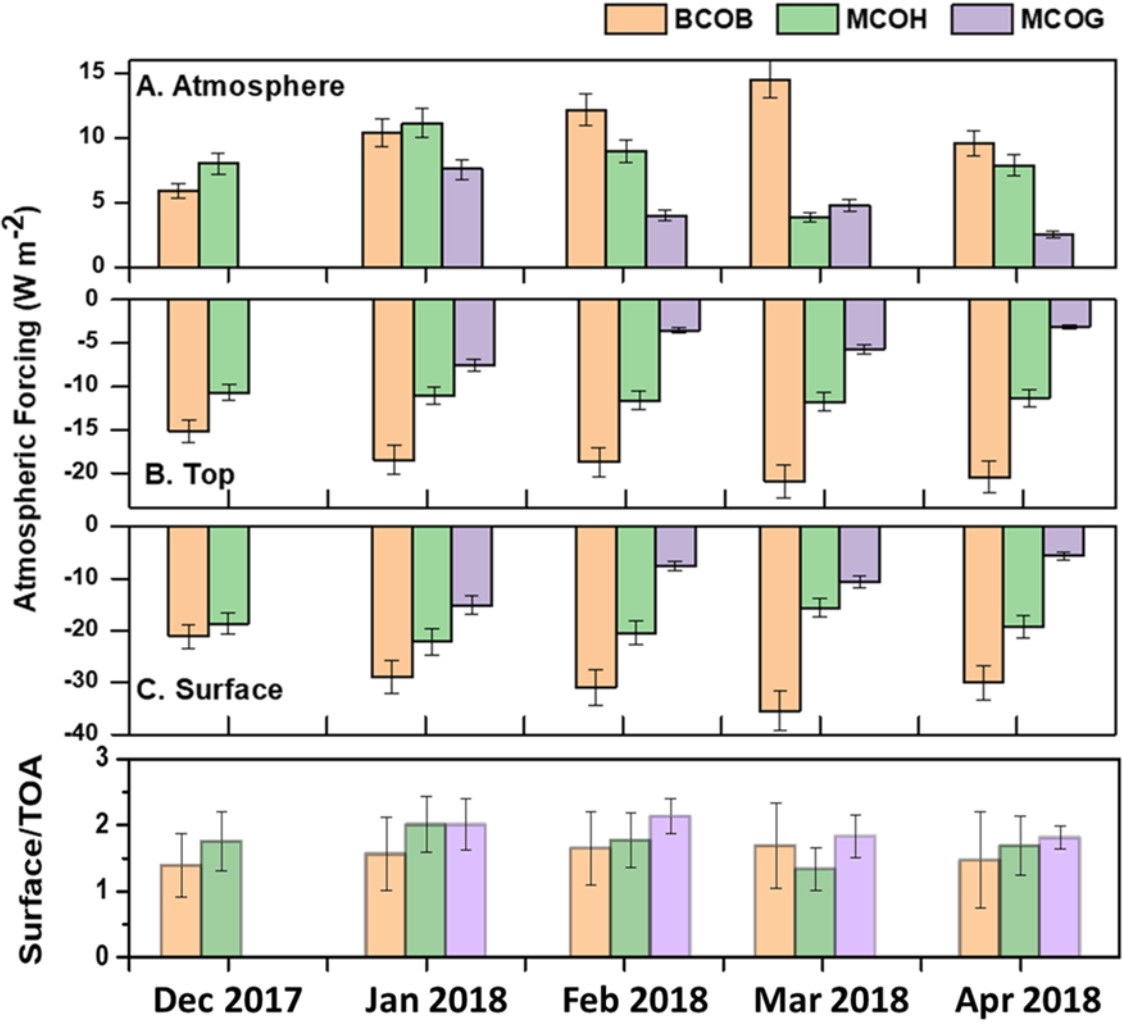

**Figure 5.** The monthly average direct aerosol radiative forcing (cloud-free atmosphere) calculated for locations of Bhola Climate Observatory-Bangladesh (BCOB), Maldives Climate Observatory-Hanimaadhoo (MCOH), and Maldives Climate Observatory-Gan (MCOG) from December 2017 to April 2018. A. Atmosphere forcing, B. Top of the atmosphere (TOA) forcing, C. Surface forcing, D. the ratio of surface forcing to the TOA.

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
