# Peer review of "Changing optical properties of Black Carbon and Brown Carbon aerosols during long-range transport from the Indo-Gangetic Plain to the equatorial Indian Ocean"

_EGUsphere, 2024_

## Author Comment (AC1)

**Author Responses and Planned Revisions to Reviewer (#1) Comments**

Budhavant et al. reports a work depicting the extent of the long-range transport of pollutants from the Indo-Gangetic Plain to Maldives via Bangladesh, and the effects of atmospheric processes happened during the transport on the properties of light absorbing species (BC and BrC). This is an interesting study with the aerosol sampling done over strategically located sites. However, there are several other studies on the similar topic using the sampling at the same locations (Bosch et al., 2014; Dasari et al., 2019; Nair et al., 2023, etc.).

We appreciate the overall positive feedback. In contrast to the cited earlier studies, this is the first study to use all three of the widely distributed receptor observatories. This enabled for the first time to constrain the optical properties of both BC and BrC, water-soluble ions, and additional parameters. Their strategic positioning allows for a comprehensive investigation of the far-reaching South Asian outflow, with the key novel insight being the doubling of the MAC of BC aerosols.

Author shall explicitly mention the novelty of the current study over the reported studies. There are some interesting results but their explanations are not satisfactory, and sometimes there are conflicting arguments while explaining the observed data. Multiple parameters (MAC678, MAC365, OC/BC, Cl/Na, [NH4], [SO4], AAE) indicate that a better data churning is required. Attribution of all observations to long-range transport and processes is not convincing. The data shall be segregated with respect to wind back trajectories and different categories shall be discussed differently with more converging inferences. There is a sufficient room for modifications before the manuscript can be considered for publication in ACP.

We will follow the reviewer's recommendations and further develop the interpretations. The three major findings of this study are:

- A near doubling of BC mass absorption cross-section (BC-MAC) from the IGP exit (at BCOB) to the Indian Ocean receptor sites, with the offered explanation backed up by ancillary data being a coating enhancement process.
- A decrease in BrC-MAC over the same over-ocean transport, with the hypothesized mechanism being the photochemical degradation of light-absorbing organic moieties.
- An enhancement of Cl/Na signals the significant presence of anthropogenic Cl at the exit of IGP, possibly affecting atmospheric oxidation.

We agree to explore segregating the data into wind sector categories for further evaluation of the results.

**Major Comments:**

L74: Why the water-insoluble BrC is ignored? There are several studies showing water-insoluble BrC is a considerable part of BrC. A comparison of water-soluble BrC alone with BC gives an incomplete and biased picture of BrC effects.

WS-BrC enhances the light absorption of BC and is evaluated also using its MAC to estimate the radiative forcing of both EC and WS-BrC. The absorption Angstrom exponent (AAE) is another optical property of BrC, indicating the wavelength dependence of light absorption, which helps extrapolate the optical properties of WS-BrC. Therefore, it is crucial to study the light absorption characteristics of both EC and WS-BrC together to quantify the relative light absorption contribution of WS-BrC to EC, which is currently lacking in the literature, especially in rural and receptor sites in South Asia. We appreciate the reviewer's suggestion and will both consider these aspects in future studies and will also here revise the current manuscript to include recognition of water-insoluble BrC.

L189-193: If this logic is true, why didn't OC come from that region? What processes can remove only OC but not $SO_4^{2-}$ and $NH_4^+$ coming from same region?

The high $SO_4^{2-}$ and $NH_4^+$ concentration at MCOH comes mainly from central and eastern India, where thermal power plants, construction industries, and petroleum refineries emit pollutants like $SO_2$ and NOx that are further oxidized in the atmosphere OC to BC decreases significantly during long-range transportation due to selective processing and/or washout of OC. We have earlier shown this system that OC has a shorter lifetime and undergoes more transformation than both BC, $SO_4$, and $NH_4$, making it a complex mixture (Budhavant et al., 2020). The differential removal rates during transportation processes have a greater impact on OC than on BC, leading to a relatively higher fraction of BC at the MCOH and MCOG sites, constituting a larger relative share of the total carbonaceous aerosols than BC (Budhavant et al., 2018, 2020).

L204-208: Discussion on Cl/Na ratio is weak and too hypothetical. The seawater ratio of Cl/Na is 1.8 (on mass basis). A dotted line with ratio 1.8 can be added to the Figure 4. At BCOB site, Cl/Na ratio is varying from 0 to 11. It shall be explained that what sources or processes are adding or removing particulate Cl from the samples. There shall be convincing supporting evidences.

We are grateful for the suggestion and will make several revisions to Figure 4, including adding a dotted line with the seawater ratio of 1.8. An enhancement of Cl may be very important. While this is not the central focus of the paper, it is of such potential significance for regional atmospheric chemistry that we feel it behoves the manuscript to bring this out. We will include more detailed descriptions to support the arguments in the revised manuscript.

L217-221: The inherent assumption in this statement is that all BC at both the receptor sites is transported from South Asia, which is not concurred by wind trajectories.

Although receptor sites MCOG and MCOG have slightly different trajectory paths, by far most of the trajectories originate from South Asia, especially during synoptic measurements. Please refer to Figure 1 and Figures S1-S5, which depict the AMBTs, fire spots, and AOD across the sampling region. While there may be other small sources on the route, such as ships and small islands, their contribution to total geographical and BC emissions is negligible. We can thus conclude that most BC emissions in the region come from South Asia. We will re-analyze this aspect during revisions and assert more clearly what the data shows.

L225-227: This is a strong but hypothetical statement. It has to be proven with other supportive measurements. Why the difference in MAC678 can't be due to different sources of BC over these sites? In fact, this inference may be more logical because OC/EC ratios are also different at those sites.

There are very few local and insignificant sources of BC over these small ocean islands compared to the heavily polluted air masses originating in the subcontinent. The decreasing OC/EC ratios are explained by higher washout of OC than EC and higher oxidation of OC than of EC, as discussed and referenced in the manuscript text. Nevertheless, we will consider the reviewer's concerns and will look to revise the text on this subject for increased clarity.

L248-250: As per the reasons given for MAC678, shouldn't MCOG have a lower MAC365 compared to MCOH?

We would like to clarify that BC and BrC have very different wavelength-dependent absorption properties and drastically differ in their involvement with other components during long-range transport. It appears that the bleaching process is becoming less significant, possibly because the most labile components have already been consumed around MCOH.

L282-284: This is quite illogical and not convincing. There are several inferences which contradicts each other.

Our description may not fully support our argument about the radiative forcing at BCOB, so we will revise the text in the revised manuscript.

Table 1: How come SO4 and NH4 conc are similar or even higher than BCOB. It is counter intuitive and warrants satisfactory justification. Concentrations of other species like OC, WSOC, NO3, K, etc. look as per expectations.

As explained earlier, the high $SO_4^{2-}$ and $NH_4^+$ concentration at MCOH comes mainly from central and eastern India, where thermal power plants, construction industries, and petroleum refineries emit pollutants like $SO_2$ and NOx that are further oxidized in the atmosphere OC to BC decreases significantly during long-range transportation due to selective processing and/or washout of OC. Nevertheless, we will consider the reviewer's concerns and will look to revise the text on this subject for increased clarity.

The increase in BC MAC678 is attributed to coating during transport from IGP to MCOH and MCOG. However, back trajectories analysis shows that winds were not from the IGP for a considerable time period. How can the higher MAC678 be justified in those samples?

In certain situations, the winds in the IGP region do not originate from within the area. Instead, they sometimes come from southern India or the Bay of Bengal. However, during winter, polluted winds from the IGP can reach the Bay of Bengal, leading to similar signals being detected over the MCOH and MCOG regions. This is especially noticeable during synoptic observations. We will elaborate on these aspects in the revised ms.

Fig. 1: 10-days air mass back trajectories are used in this paper, which is in contrast to most of the studies which are using 5 or 7 day back trajectories. The reason of using 10 day back trajectory shall be explained.

The lifespan of BC is influenced by wet and dry deposition, with fine-mode aerosols tending to persist for longer periods. Factors such as humidity, wind speed, temperature, and mixing state also affect its lifespan, which typically ranges from one to two weeks. Based on our earlier studies on this Indian Ocean receptor, it has become clear that anthropogenic aerosols at this time may have quite a long residence time, even surpassing two weeks (e.g., Budhavant et al., 2020, 2023). This is the key motivation for having longer BT times than 5-7 days. This study examines air mass back trajectories to identify potential sources of BC and other aerosol components at the MCOG station, which is even further from MCOH. Given this long-distance travel, the focus on the dry season (increasing longevity), the fact that BC has longer lifetimes than other aerosols, and the experiences from earlier studies, a BT time horizon of ten days is deemed appropriate. We will in the revised manuscript elaborate on this motivation.

Fig.2: This data shall also be plotted in different ways. Samples with similar BC fractions are showing quite different MAC678. It shall be explained.

We will revise the manuscript to acknowledge and explain that similar BC fractions may display different $MAC_{678}$ due to differences in, e.g., coating and internal mixing.

Fig. 3: As per this plot, WIOC is the dominant fraction of OC. A significant part of this WIOC could be BrC, which is not measured, reflecting limitation of this work. Further, in many samples with low WSOC fraction, MAC365 is quite high. Possible reasons shall be discussed.

This point is already made above. While this and earlier studies are showing that WSOC is the dominant fraction of OC and an analytically tangible fraction that contains light-absorbing moieties, the manuscript will be revised on this aspect to more explicitly acknowledge that there is also WIOC.

Fig. 4: In many of the samples collected over MCOH and MCOG, OC/BC looks close to 1, which is not normal. How is it inferred?

It is correct that the ratios of OC/BC are relatively low compared to other studies conducted in mainland South Asia, except in certain urban cities such as Dhaka and Chennai. Nevertheless, the OC/EC ratios reported for MCOH in many other earlier studies align with the current study's findings on this ratio. This suggests decreasing OC/BC between IGP exit and after transportation over the ocean, indicating selective washout and bleaching reactions of organic carbon, as discussed in detail in the m (page 6, line 183-185).

It would be better to plot WSOC/OC rather than WSOC/TC because WSOC/OC ratio can be better interpreted.

We will implement this change suggested by the reviewer in Figure 4 during revisions.

Fig.5: As the major focus of this paper is on BC and BrC, it would be appropriate to calculate RF for BC and BrC, and their contribution to total aerosols RF.

Thank you for your suggestion, but it is currently beyond the scope of our study. However, we will consider it for future studies.

Fig.S2-S4: Why 10 days and not 5 or 7 days? Why at 50 m only? It would be better to add a few higher altitudes relevant for long-range transport.

Please refer to our response to the back trajectories of Figure 1 a few comments earlier. The sampling towers of the three atmospheric observatories have an altitude below 20 meters on their islands. We hence used a height of 50 meters to compute the air mass back trajectories for the sampled boundary layer.

Fig, S5: AOD data appears to follow expected trends, unlike chemical data.

We believe both AOD and chemical data can be understood, as discussed in the manuscript.

Table S1: $SO_4$, $NH_4$, nss-Ca, nss-Mg are not correlated with any species, why? Where is Na and Cl? Major ions data (absolute concentrations) shall also be given here.

Nss-$SO_4$ comes mainly from industries, coal power plants, and ships; $NH_4$ comes mainly from agriculture, including animal husbandry and $NH_3$-based fertilizer application, whereas nss-Ca and nss-Mg are primarily sourced from soil. Thus, it is not expected that all these are correlated with each other. While certain correlations do exist between some measured ions, such as nss-$SO_4$ and $NO_3$, nss-Ca, and nss-Mg, they were not explicitly detailed in the manuscript since the paper primarily focuses on optical features. Additionally, we will include Na and Cl in Tables S1-S3 and provide their concentrations in Table 2.

Table S2: Why do NO3, Ca, and Mg are not correlating with any other species?

Please refer to our previous response to the comment above.

Table S3: Why does K not correlating with any other species?

Please refer to the response to the earlier comment.

**Minor Comments:**

L53-56: add appropriate references from this region.

We will add more references.

L120: Eq 3: How was the babs measured? Why at 365 nm only?

The absorption spectrum of water extracts was measured using a Hitachi absorption spectrophotometer-2010, in the range of 190 to 1200 nm (as described in the Methods section; lines 118-126). BrC is an organic aerosol moiety that absorbs light of shorter wavelengths. It is mainly absorbed in the ultraviolet and near-visible wavelengths, which gives it a brownish or yellowish appearance. To accurately measure the levels of BrC present in particles, the absorption coefficient between 360 and 370 nm (average of 365 nm) is commonly used and allows for inter-comparison of BrC between studies

L255-257: There are numerous recent studies and MAC365 shall be compared with the recent studies.

We will compare our results with even more earlier reports of this property in the revised version of the manuscript.

L262-264: mention the wavelength range used for AAE calculation for the better clarity for readers.

To prevent any potential interference from light-absorbing solutes such as ammonium nitrate, sodium nitrate, and nitrate ions, which have absorption peaks near 308, 298, and 302 nm, respectively, the AAE was based on the 330-400 nm range, as is common praxis. This is already mentioned in the manuscript in the earlier section, lines 125-126.

---

## Author Comment (AC2)

**Author Responses and Planned Revisions to Reviewer (#2) Comments**

The Budhavant manuscript describes the results of SAPOEX campaign at three sites in South Asia. There is potential with this dataset, but the authors need to do more to connect across the results in each section (e.g. back trajectory analysis, aerosol composition, optical properties, radiative forcing). For example, the absolute concentrations of aerosol composition were only presented as campaign averages, which made it difficult to compare to the aerosol radiative forcing which was presented and discussed as an average and by month. There was often an over-simplification of the results, for example, the assertion that the MCOH site received transported air masses from the BCOB site: this did not seem to be always the case so it added confusion when discussing composition and aging. Overall the manuscript needs more refining of focus and connection among the different sections. Even in the introduction, the discussion of source and processing impacts on BC and BrC was over-simplified and lacking in precision.

We appreciate this constructive feedback and concrete suggestions to enhance our manuscript. We are committed to making these necessary revisions and providing supplementary information to support our findings. We will scrutinize for opportunities to relate the parts closer to each other and synthesize the overall findings.

**Detailed comments:**

Ln 25: can this be better linked in the abstract to the aerosol optical properties?

Yes. The revised manuscript will link the similar pattern between Br-MAC and total radiative forcing to aerosol optical properties.

Ln 46: this is an awkward phrase here. please edit

We will edit this text bit for clarity.

Ln 47: is this referring specifically to this region? this is undoubtedly true for BC, but WSOC may have other sources? e.g. biogenic and SOA?

Thank you for bringing this additional dimension up. We acknowledge that WSOC may have additional sources, such as biogenic SOA, and we will make the necessary changes to the manuscript.

Ln 47 – 56: this paragraph is difficult to follow as written.  the authors need to clarify their purpose here. There is some confusion as they are trying to simultaneously discuss BC and WS-BrC. It doesn't really work and needs editing for clarity.

We recognize that our initial is blurred.  We will edit thoroughly, likely breaking the paragraph up into two separate ones for the two aspects.

Ln 49: This sentence is awkward and the logical transition here is unclear

We will revisit to address the issue.

Ln 65: edit for clarity

We will edit for clarity.

Ln 88: What is the particle size here? TSP? PM2.5

We have used $PM_{2.5}$ samples. This will be explicitly mentioned here.

Ln 111: are there any concerns about the high loading on these filters? typically filter-based photometers limit the filter loading to that which corresponds to a 50% transmission. the filters collected for this offline analysis would not have their loading limited by light transmission. i understand that this correction is intended to address the filter loading, but these correction schemes were originally designed for online instruments which have a filter advancement/change at a set transmission threshold. can this concern be addressed?

Yes, the filter loading is a concern in this highly polluted air (especially for BCOB). We have invested substantial consideration into any effects of this and have thus previously reported and discussed filter-loading corrections and these are cited in this manuscript (e.g., Budhavant et al., 2020).

Ln 180-182: its not clear how these facts are relevant here. please remove or expand the discussion to make this more clear.

We will revisit this part and remove any irrelevant parts, and add more supporting explanations as needed.

Ln 191-195: what about biogenic SO4? do you have a constraint on the possible marine contribution that goes beyond sea salt? Additionally, this rationale of sources from central and east India is a bit confusing as the BTs indicate that air masses predominantly leave the Indian subcontinent near BCOB or from west India before traveling to MCOH. If the aerosol composition from the west side of India is markedly different (e.g. higher SO4 fraction) than the IGP and BCOB, than the aging discussion of MCOH representing aged BCOB aerosol needs to be more refined.

A previous study at MCOH found that DMS contributes only up to 3% to nss-SO4 in polluted air (Granat et al., 2010). We will evaluate the SO4/BC load for samples coming from west India vs from the Bay of Bengal. The "synoptic" comparison will be focused on winds coming from the IGP and southern India. The revised manuscript will be updated to this effect.

Ln 208: Is the EC supposed to be BC?

We confirm that this is BC.

Ln 217-223: I'm still stuck on the SO4 discussion. the provenance of the SO4 seems very relevant in determining if these 3 sites do represent different ages of the same air mass. as

briefly mentioned in this section, increased SO4 would also seem to be very relevant for the coatings question. however some of the previous discussion of the loss of water soluble fraction during aging and transport (WSOC) seems to conflict with this. certainly the increase in SO4 at MCOH, absolute concentration as well as an extreme increase in relative contribution, is very relevant for coating of BC and internal mixture of BC and SO4 aerosol. i would like to see more discussion of this potentially conflicting observations between wsoc and so4, and using the three sites as steps in the aging process of one air mass. If the rationale for higher SO4 at MCOH is a shift in geographic source region, than the discussion of aging of two different aerosol systems needs to be included.

We agree with this thinking. We will return to these aspects and seek to both clarify and provide additional information by incorporating two distinct aerosol systems.

Ln 265-267: are these differences in AAE signficant? the std deviation is relatively high.

We have noted the differences mentioned, but they do not seem significant. Hence, we will proceed with making the necessary changes.

Ln 282-284: can the authors discuss why the atmospheric forcing was higher at MCOH while the surface concentrations for summed species was lower? it is also a bit difficult to interpret the relationship between the aerosol radiative forcing and the rest of the surface aerosol discussion as the time scales are not well aligned for aerosol concentrations and the ARF. overall, i'd like to see better connections among the sections of the discussion.

In general, MCOH has lower radiative forcing than BCOB. However, during January, MCOH had slightly higher atmospheric forcing due to outflow from IGP, as shown in Figures 5, 2, and 3. We will provide more clarity in the revised version. We will also seek to connect/synthesize the findings in the different sections further.

Table 1: this AAE is calculated off a very narrow range in wavelength. is 400 nm the longest wavelength measured here? What are the potential shortcomings of reporting AAE for such a narrow range in wavelength? Also, it is difficult to assess the trends in the ambient concentrations when only the averages for the entire period are reported. It is fine to present the ratios in the figures, but useful to also be able to see changes in the absolute concentrations as well.

We have measured the wavelength range from 190 to 1200 nm, yet, as elaorated above in response to reviewer 1, the AAE is customarily reported for the range of 330 to 400 nm. One reason is to prevent any potential interference from light-absorbing solutes such as ammonium nitrate, sodium nitrate, and nitrate ions, which have absorption peaks near 308, 298, and 302 nm. Further AAE is more dependent on linear ratios for shorter wavelengths, while for longer wavelengths, the correlation is weaker. We plan to revise to include these motivations in the additional information for Table 2.

---

## Author Response (AR1)

**Author Responses to Reviewer's Comments and Planned Revisions**

Manuscript Ref No: egusphere-2024-104

Manuscript: '**Changing optical properties of Black Carbon and Brown Carbon aerosols during long-range transport from the Indo-Gangetic Plain to the equatorial Indian Ocean**' by K. Budhavant, M. R. Manoj, H.R.C.R. Nair, S. M. Gaita, H. Holmstrand, A. Salam, A. Muslim, S. K. Satheesh, Ö. Gustafsson.

We are grateful for the overall positive feedback from both reviewers. Their comments and suggestions have helped us to further improve our manuscript, both with respect to its scientific content and its structure. Revisions and our responses to each review comment are clearly marked in blue below. The place of these revisions in the revised manuscript is also indicated with the line number (of the revised manuscript). We have also made some additional revisions to the manuscript to further increase clarity and relate to additional/new literature.

**Reviewer (#1) comments**

Budhavant et al. reports a work depicting the extent of the long-range transport of pollutants from the Indo-Gangetic Plain to Maldives via Bangladesh, and the effects of atmospheric processes happened during the transport on the properties of light absorbing species (BC and BrC). This is an interesting study with the aerosol sampling done over strategically located sites. However, there are several other studies on the similar topic using the sampling at the same locations (Bosch et al., 2014; Dasari et al., 2019; Nair et al., 2023, etc.).

We appreciate the overall positive feedback. In contrast to the cited earlier studies, this is the first study to use all three widely distributed receptor observatories of the South Asian outflow. This study also enabled for the first time to constrain the optical properties of both BC and BrC, interpreted along with ancillary data also on water-soluble ions and additional parameters. The strategic positioning of the observatories allows for a comprehensive investigation of the far-reaching South Asian outflow, with perhaps the key novel insight being the observational constraint of a near-doubling of the Mass Absorption Coefficient (MAC) of BC aerosols from the continental exit until arrival to the northern Indian Ocean receptors sites.

Author shall explicitly mention the novelty of the current study over the reported studies. There are some interesting results but their explanations are not satisfactory, and sometimes there are conflicting arguments while explaining the observed data. Multiple parameters (MAC678, MAC365, OC/BC, Cl/Na, [NH4], [SO4], AAE) indicate that a better data churning is required. Attribution of all observations to long-range transport and processes is not convincing. The data shall be segregated with respect to wind back trajectories and different categories shall be

discussed differently with more converging inferences. There is a sufficient room for modifications before the manuscript can be considered for publication in ACP.

We have followed the reviewer's recommendations and further developed the interpretations. The three major findings of this study are:

- A near doubling (80%) of BC mass absorption cross-section (BC-MAC) from the IGP exit (at BCOB) to the Indian Ocean receptor sites. This is hypothesized to reflect mainly a scavenging fractionation of the BC that preferentially retains smaller BC particles with intrinsically higher MAC. This has large implications for estimating the radiative/climate effects of BC aerosols over South Asia.
- A decrease in BrC-MAC over the same over-ocean transport, with the hypothesized mechanism for this being the photochemical degradation of light-absorbing organic moieties.
- An enhancement of $Cl^-/Na^+$, which signals a significant presence of anthropogenic Cl at the exit of IGP, possibly affecting atmospheric oxidation.

We have added a more detailed discussion about air mass back trajectories to gain better clarity. Page 5, lines 158-162; page 6, lines 179-184.

**Major Comments:**

L74: Why the water-insoluble BrC is ignored? There are several studies showing water-insoluble BrC is a considerable part of BrC. A comparison of water-soluble BrC alone with BC gives an incomplete and biased picture of BrC effects.

The WIOC is not analytically attainable and thus very hard to constrain observationally. Hence, the BrC part of this study (a minor part of this study, BC is the main scope) focuses on WS-BrC, which allows comparison w BC and with the many earlier studies of WS-BrC.

L189-193: If this logic is true, why didn't OC come from that region? What processes can remove only OC but not SO42- and NH4+ coming from same region?

The high $SO_4^{2-}$ and $NH_4^+$ concentration at MCOH comes mainly from central and eastern India, where thermal power plants, construction industries, and petroleum refineries emit pollutants like $SO_2$ and NOx that are further oxidized in the atmosphere. The OC-to-BC decreases significantly during long-range transportation due to selective processing and/or washout of OC. We have earlier shown for this system that OC has a shorter lifetime and undergoes more transformation than BC, $SO_4^{2-}$, and $NH_4^+$ (Budhavant et al., 2020). The differential removal rates during long-range transport have a greater impact on OC than on BC, $SO_4^{2-}$ and $NH_4^+$, leading to a relatively higher fraction of BC at the MCOH and MCOG sites, thus constituting a larger relative share of the total carbonaceous aerosols than OC (Budhavant et al., 2018, 2020). Page 6, lines 193-197; Page 8, lines 255-263.

L204-208: Discussion on Cl/Na ratio is weak and too hypothetical. The seawater ratio of Cl/Na is 1.8 (on mass basis). A dotted line with ratio 1.8 can be added to the Figure 4. At BCOB site, Cl/Na ratio is varying from 0 to 11. It shall be explained that what sources or processes are adding or removing particulate Cl from the samples. There shall be convincing supporting evidences.

We are grateful for the suggestion and have made several revisions to Figure 4, including adding a dotted line with the seawater ratio of 1.8. Enhanced Cl$^-$ may have very important consequences for atmospheric chemistry. While this is not the central focus of the paper, it is of such potential significance for regional atmospheric chemistry that we feel it behoves the manuscript to bring this out. We have included more detailed descriptions to support the arguments in the revised manuscript. Page 8, lines 238-248.

"*Marine aerosol often experiences chlorine depletion, and releasing gas-phase HCl from particles can impact aerosol acidity and the concentration of water-soluble ions. However, once these particles enter the atmosphere, they become exposed to various pollutants, leading to the loss of particulate chlorine into the gaseous phase. This loss of chlorine is typically attributed to ion exchange reactions with atmospheric acids like $SO_2$, $H_2SO_4$, and $HNO_3$, which result in the formation of sulfates and nitrates, as well as the degassing of HCl (Orsini et al., 1986; Brimblecombe and Clegg, 1988; Haslett et al., 2023). Other pathways lead to the loss of particulate chlorine, such as interaction with NO: $N_2O_5$, HOBr, and $O_3$, as well as the release of NOCl, HONO, $ClNO_2$, $Cl_2$, and BrC (Vogt et al., 1996; Behnke and Zetzsch, 1989; Haslett et al., 2023). These pathways can have significant implications for marine tropospheric chemistry and the polluted coastal atmosphere due to the creation of photochemically active halogenated gaseous compounds. The study found significant anthropogenic chloride emissions from human activities, which can affect the oxidation capacity of polluted air.*"

L217-221: The inherent assumption in this statement is that all BC at both the receptor sites is transported from South Asia, which is not concurred by wind trajectories.

Although receptor sites MCOH and MCOG have slightly different trajectory paths, by far most of the trajectories originate from South Asia, especially during these wintertime measurements. Please refer to Figure 1 and Figures S1-S5, which depict the AMBTs, fire spots, and AOD across the sampling region. While there may be other small sources along the route, such as ships and small islands, their contribution to total geographical and BC emissions is negligible. We can thus conclude that most BC emissions in the region come from South Asia. We have re-analyzed this aspect during revisions and elaborated more clearly on what the data shows. Page 6, lines 179-182; Page 8, lines 260-262.

"*The MCOH and MCOG are situated in the northern Indian Ocean and thus intercept long-range pollutant emissions from South Asia, including the IGP, the western part of India, and the Indian Ocean (Figure S3 and Figure S4). Occasionally, the winds in the IGP sector come from southern India or the Bay of Bengal. However, during winter, polluted winds from the*

*IGP can reach the Bay of Bengal, leading to similar signals being detected over the MCOH and MCOG regions. This is particularly noticeable during synoptic observations."*

*"It is worth noting that there could be some minor sources of emissions along the route to the receptor sites, like ships and small islands. However, their impact on the overall regional loading of BC is insignificant. Therefore, we can infer that most BC loading over the northern Indian Ocean originates from high-emission source areas in South Asia."*

L225-227: This is a strong but hypothetical statement. It has to be proven with other supportive measurements. Why the difference in MAC678 can't be due to different sources of BC over these sites? In fact, this inference may be more logical because OC/EC ratios are also different at those sites.

There are very few local and then insignificant sources of BC over these small ocean islands compared to the heavily polluted air masses originating in the subcontinent and reaching these locations. The decreasing OC/EC ratios are explained by higher washout of OC than EC and higher oxidation of OC than of EC, as discussed and referenced in the manuscript text. Nevertheless, we have considered the reviewer's concerns and revised the text for increased clarity. More details are given in our answer to the earlier comment.

L248-250: As per the reasons given for MAC678, shouldn't MCOG have a lower MAC365 compared to MCOH?

We would like to clarify that BC and BrC have very different wavelength-dependent absorption properties and drastically differ in their involvement with other components during long-range transport. It appears that the bleaching process of BrC is becoming less significant with further transport to equatorial MCOG, possibly because most labile components have already been consumed by the time the air parcels pass by MCOH.

L282-284: This is quite illogical and not convincing. There are several inferences which contradicts each other.

Our description may not fully support our argument about the radiative forcing at BCOB, so we have revised the text in the revised manuscript. Page 10-11, lines 337-344 and 346-350.

*"During winter, the IGP experiences a significant increase in aerosol loading, mainly carbon aerosols resulting from fossil fuel and biofuel combustion (Gustafsson et al., 2009; Kaskaoutis et al., 2014; Dasari et al., 2020). As spring progresses, dust becomes the dominant aerosol in the northwestern region of India and arid areas of southwest Asia (Kaskaoutis et al., 2014; Singh et al., 2014; Dumka et al., 2023). At the same time, significant agricultural burning in*

*Southeast Asia results in significantly elevated concentrations of carbonaceous aerosols (Kaskaoutis et al., 2014; Budhavant et al., 2015; Bikina et al., 2019). Also, Himalayan forest fires and wheat residue burning in the IGP contribute to the aerosol burden during spring (Gautam et al., 2007; Bikina et al., 2019). BCOB experienced a high atmospheric forcing in March, particularly in the outflow region of the IGP (Figure 5). In January, MCOH experienced slightly higher atmospheric forcing (11.2 $Wm^{-2}$) than BCOB (10.4 $Wm^{-2}$). The findings are consistent with our earlier study conducted at MCOH, which showed that anthropogenic aerosols, such as BC, BC, $nss\text{-}K^+$, $nss\text{-}SO_4^{2-}$, and $NH_4^+$, were predominantly in the fine mode (70-95%) and particularly observed in the air masses coming from IGP during the period (Budhavant et al., 2018). Therefore, it is crucial to address this issue and take appropriate measures to reduce the amount of anthropogenic aerosol loading*."

Table 1: How come SO4 and NH4 conc are similar or even higher than BCOB. It is counter intuitive and warrants satisfactory justification. Concentrations of other species like OC, WSOC, NO3, K, etc. look as per expectations.

As explained earlier, the relatively high $SO_4^{2-}$ and $NH_4^+$ concentration at MCOH comes mainly from central and eastern India, where thermal power plants, construction industries, and petroleum refineries emit pollutants like $SO_2$ and NOx that are further oxidized in the atmosphere. The OC-to-BC ratio decreases significantly during long-range transportation due to selective processing and/or washout of OC. Nevertheless, in consideration of the reviewer's concerns, the text was revised on this subject for increased clarity.  Page 7, lines 223-224.

"*Furthermore, a previous study at MCOH found that dimethyl sulfide (DMS) contributes only up to 3% to $nss\text{-}SO_4^{2-}$ in polluted air (Granat et al., 2010).*"

The increase in BC MAC678 is attributed to coating during transport from IGP to MCOH and MCOG. However, back trajectories analysis shows that winds were not from the IGP for a considerable time period. How can the higher MAC678 be justified in those samples?

In certain situations, the winds in the IGP region do not originate from within the area. Instead, they sometimes come from southern India or the Bay of Bengal (Figure S3-S4). However, during winter, polluted winds from the IGP can reach the Bay of Bengal, leading to similar signals being detected over the MCOH and MCOG regions. This is especially noticeable during synoptic observations.  We have already elaborated on these aspects in the earlier comments. The revised manuscript now has an expanded discussion on the underlying causes of the increasing BC $MAC_{678}$ during the long-range transport, including the likely preferential removal of larger BC fractions so that finer BC with presumably higher $MAC_{678}$ is preferentially retained.  Page 9, lines 272-282.

Fig. 1: 10-days air mass back trajectories are used in this paper, which is in contrast to most of the studies which are using 5 or 7 day back trajectories. The reason of using 10 day back trajectory shall be explained.

The lifespan of BC is influenced by wet and dry deposition, with fine-mode aerosols tending to persist for longer periods. Factors such as humidity, wind speed, temperature, and mixing state also affect its lifespan, typically from one to two weeks. Based on our earlier studies on this Indian Ocean receptor, it has become clear that anthropogenic aerosols at this time may have quite a long residence time, even surpassing two weeks (e.g., Budhavant et al., 2020, 2023). This is the key motivation for having longer BT times than 5-7 days. This study examines air mass back trajectories to identify potential sources of BC and other aerosol components at the MCOG station, which is even further from MCOH. Given this long-distance travel, the focus on the dry season (increasing longevity), BC has longer lifetimes than other aerosols, and the experiences from earlier studies, a BT time horizon of ten days is deemed appropriate. In the revised manuscript, we have elaborated on this motivation. Page 5, lines 158-162.

"*This study examines air mass back trajectories to identify potential sources of BC and other aerosol components arriving at the MCOG station, which is even further away than MCOH from source regions. Given this long-distance travel, the focus on the dry season (increasing longevity), that BC has longer lifetimes than other aerosols, and the experiences from earlier studies (Budhavant et al., 2020, 2023), a BT time horizon of ten days was selected as appropriate.*"

Fig.2: This data shall also be plotted in different ways. Samples with similar BC fractions are showing quite different MAC678. It shall be explained.

We have revised the manuscript to acknowledge and explain that similar BC fractions may display different $MAC_{678}$ due to differences in, e.g., coating, sizes and internal mixing. Page 9, lines 272-283.

Fig. 3: As per this plot, WIOC is the dominant fraction of OC. A significant part of this WIOC could be BrC, which is not measured, reflecting limitation of this work. Further, in many samples with low WSOC fraction, MAC365 is quite high. Possible reasons shall be discussed.

This point is already made and addressed above. While this and earlier studies show that WSOC is the dominant fraction of OC and an analytically tangible fraction that contains light-absorbing moieties, the manuscript has been revised to acknowledge more explicitly that there is also WIOC. Page 7, lines 202-203.

Fig. 4: In many of the samples collected over MCOH and MCOG, OC/BC looks close to 1, which is not normal. How is it inferred?

It is correct that the ratios of OC/BC are relatively low this far out over the Indian Ocean compared to in studies conducted in mainland South Asia. Nevertheless, the OC/EC ratios reported for MCOH in many other earlier studies align with the current study's findings on this

ratio (Budhavant et al., 2018, 2020; Nair et al., 2024). This suggests decreasing OC/BC between IGP exit and after long-range transport over the ocean, indicating selective washout and bleaching reactions of organic carbon, as discussed in detail in the manuscript (e.g., page 6, lines 192-201).

It would be better to plot WSOC/OC rather than WSOC/TC because WSOC/OC ratio can be better interpreted.

We have implemented this change suggested by the reviewer in Figure 4.

[Figure]

*Figure 4. Time series of the ratio of measured chemical species OC/EC (panel A), SO4/BC (panel B), WSOC/BC (panel C), and Cl/Na (panel D, seawater ratio 1.8, dotted line) over three receptor sites in South Asia: Bhola Climate Observatory-Bangladesh (BCOB), Maldives Climate Observatory-Hanimaadhoo (MCOH), and Maldives Climate Observatory-Gan (MCOG).*

Fig.5: As the major focus of this paper is on BC and BrC, it would be appropriate to calculate RF for BC and BrC, and their contribution to total aerosols RF.

Thank you for your suggestion, but it is well beyond the scope of our study. However, we will consider it for future studies.

Fig.S2-S4: Why 10 days and not 5 or 7 days? Why at 50 m only? It would be better to add a few higher altitudes relevant for long-range transport.

Please refer to our response to the back trajectories of Figure 1, which was a few comments earlier. The sampling towers of the three atmospheric observatories have an altitude below 20 meters on the islands. We, therefore, used a height of no more than 50 meters to compute the air mass back trajectories for the sampled boundary layer.

Fig, S5: AOD data appears to follow expected trends, unlike chemical data.

We believe both AOD and chemical data can be understood, as discussed in the manuscript.

Table S1: SO4, NH4, nss-Ca, nss-Mg are not correlated with any species, why? Where is Na and Cl? Major ions data (absolute concentrations) shall also be given here.

Nss-$SO_4^{2-}$ comes mainly from industries, coal power plants, and ships; $NH_4^+$ comes mainly from agriculture, including animal husbandry and $NH_3$-based fertilizer application, whereas nss-$Ca^{2+}$ and nss-$Mg^{2+}$ are primarily sourced from soil. Thus, since these components come from different sources, they are not expected to be correlated with each other. While certain correlations do exist between some measured ions, such as nss-$SO_4^{2-}$ and $NO_3^-$, nss-$Ca^{2+}$, and nss-$Mg^{2+}$, they were not explicitly detailed in the manuscript since the paper primarily focuses on optical features of carbon aerosols. Nevertheless, we have now revised to include $Na^+$ and $Cl^-$ in Tables S1-S3 and provide their concentrations in Table 2.

**Table S1. Matrix of correlation coefficients (r) for the components measured at BCOB station. Correlations coefficients higher than 0.7 are highlighted in bold.**

|  | OC | EC | WSOC | WIOC | nss-SO4 | NO3 | nss-K | nss-Ca | nss-Mg | NH4 | Na | Cl |
|---|---|---|---|---|---|---|---|---|---|---|---|---|
| **OC** | **1.00** | **0.85** | **0.95** | **0.99** | 0.11 | **0.80** | **0.85** | -0.36 | -0.44 | -0.23 | 0.22 | **0.76** |
| **EC** | **0.85** | **1.00** | **0.79** | **0.85** | -0.05 | 0.55 | **0.79** | -0.46 | -0.54 | -0.22 | 0.08 | **0.74** |
| **WSOC** | **0.95** | **0.79** | **1.00** | **0.89** | 0.10 | **0.76** | **0.78** | -0.41 | -0.47 | -0.19 | 0.17 | **0.69** |
| **WIOC** | **0.99** | **0.85** | **0.89** | **1.00** | 0.11 | **0.78** | **0.85** | -0.32 | -0.42 | -0.24 | 0.23 | **0.77** |
| **nss-SO4** | 0.11 | -0.05 | 0.10 | 0.11 | **1.00** | 0.40 | 0.28 | 0.21 | 0.09 | -0.03 | 0.14 | 0.22 |
| **NO3** | **0.80** | 0.55 | **0.76** | **0.78** | 0.40 | **1.00** | **0.77** | -0.12 | -0.23 | -0.28 | 0.35 | **0.73** |
| **nss-K** | **0.85** | **0.79** | **0.78** | **0.85** | 0.28 | **0.77** | **1.00** | -0.38 | -0.43 | -0.27 | 0.30 | **0.82** |
| **nss-Ca** | -0.36 | -0.46 | -0.41 | -0.32 | 0.21 | -0.12 | -0.38 | **1.00** | 0.53 | 0.40 | 0.17 | -0.14 |
| **nss-Mg** | -0.44 | -0.54 | -0.47 | -0.42 | 0.09 | -0.23 | -0.43 | 0.53 | **1.00** | 0.05 | 0.63 | -0.20 |
| **NH4** | -0.23 | -0.22 | -0.19 | -0.24 | -0.03 | -0.28 | -0.27 | 0.40 | 0.05 | **1.00** | -0.17 | -0.27 |
| **Na** | 0.22 | 0.08 | 0.17 | 0.23 | 0.14 | 0.35 | 0.30 | 0.17 | 0.63 | -0.17 | **1.00** | 0.46 |
| **Cl** | **0.76** | **0.74** | **0.68** | **0.77** | 0.22 | **0.73** | **0.82** | -0.14 | -0.20 | -0.27 | 0.46 | **1.00** |

**Table S2. Matrix of correlation coefficients (r) for the components in PM$_{2.5}$ measured at MCOH. Correlations coefficients higher than 0.7 are highlighted in bold.**

| | OC | EC | WSOC | WIOC | nss-SO$_4$ | NO$_3$ | nss-K | nss-Ca | nss-Mg | NH$_4$ | Na | Cl |
|---|---|---|---|---|---|---|---|---|---|---|---|---|
| **OC** | **1.00** | **0.74** | **0.72** | **0.93** | 0.60 | 0.21 | 0.55 | 0.33 | -0.13 | 0.60 | 0.27 | 0.11 |
| **EC** | **0.74** | **1.00** | 0.61 | 0.64 | **0.78** | 0.40 | **0.74** | 0.25 | 0.04 | **0.78** | 0.67 | 0.47 |
| **WSOC** | **0.72** | 0.61 | **1.00** | 0.60 | **0.82** | 0.32 | **0.77** | 0.29 | -0.08 | **0.83** | 0.58 | 0.52 |
| **WIOC** | **0.93** | 0.64 | 0.60 | **1.00** | 0.48 | -0.01 | 0.46 | 0.32 | -0.04 | 0.49 | 0.67 | 0.50 |
| **nss-SO$_4$** | 0.60 | **0.78** | **0.82** | 0.48 | **1.00** | 0.38 | **0.94** | 0.21 | 0.05 | **0.99** | **0.72** | **0.76** |
| **NO$_3$** | 0.21 | 0.40 | 0.32 | -0.01 | 0.38 | **1.00** | 0.32 | 0.12 | 0.01 | 0.38 | **0.84** | **0.80** |
| **nss-K** | 0.55 | **0.74** | **0.77** | 0.46 | **0.94** | 0.32 | **1.00** | 0.15 | 0.08 | **0.95** | **0.70** | 0.59 |
| **nss-Ca** | 0.33 | 0.25 | 0.29 | 0.32 | 0.21 | 0.12 | 0.15 | **1.00** | 0.23 | 0.19 | 0.10 | -0.20 |
| **nss-Mg** | -0.13 | 0.04 | -0.08 | -0.04 | 0.05 | 0.01 | 0.08 | 0.23 | **1.00** | 0.05 | -0.04 | -0.47 |
| **NH$_4$** | 0.60 | **0.78** | **0.83** | 0.49 | **0.99** | 0.38 | **0.95** | 0.19 | 0.05 | **1.00** | **0.72** | 0.49 |
| **Na** | 0.27 | 0.67 | 0.58 | 0.67 | **0.72** | **0.84** | **0.70** | 0.10 | -0.04 | **0.72** | **1.00** | 0.59 |
| **Cl** | 0.11 | 0.47 | 0.52 | 0.50 | **0.76** | **0.80** | 0.59 | -0.20 | -0.47 | 0.49 | 0.59 | **1.00** |

**Table S3. Matrix of correlation coefficients (r) for the components measured at MCOG station. Correlations coefficients higher than 0.7 are highlighted in bold.**

| | OC | EC | WSOC | WIOC | nss-SO$_4$ | NO$_3$ | nss-K | nss-Ca | nss-Mg | NH$_4$ | Na | Cl |
|---|---|---|---|---|---|---|---|---|---|---|---|---|
| **OC** | 1.00 | **0.83** | **0.80** | **0.99** | 0.55 | **0.68** | 0.26 | 0.56 | 0.39 | 0.43 | 0.53 | 0.48 |
| **EC** | **0.83** | **1.00** | **0.67** | **0.83** | **0.79** | 0.50 | 0.10 | 0.39 | 0.27 | 0.63 | 0.53 | 0.35 |
| **WSOC** | **0.80** | **0.67** | **1.00** | **0.73** | 0.50 | 0.53 | 0.02 | 0.44 | 0.26 | 0.43 | 0.43 | 0.38 |
| **WIOC** | **0.99** | **0.83** | **0.73** | **1.00** | 0.53 | **0.68** | 0.30 | 0.56 | 0.39 | 0.41 | 0.52 | 0.47 |
| **nss-SO$_4$** | 0.55 | **0.79** | 0.50 | 0.53 | **1.00** | 0.30 | 0.17 | 0.29 | 0.22 | **0.78** | 0.54 | 0.04 |
| **NO$_3$** | **0.68** | 0.50 | 0.53 | **0.68** | 0.30 | **1.00** | 0.24 | **0.92** | **0.85** | 0.00 | **0.74** | **0.80** |
| **nss-K** | 0.26 | 0.10 | 0.02 | 0.30 | 0.17 | 0.24 | **1.00** | 0.31 | 0.47 | 0.24 | 0.04 | 0.14 |
| **nss-Ca** | 0.56 | 0.39 | 0.44 | 0.56 | 0.29 | **0.92** | 0.31 | **1.00** | 0.91 | 0.04 | 0.68 | **0.70** |
| **nss-Mg** | 0.39 | 0.27 | 0.26 | 0.39 | 0.22 | **0.85** | 0.47 | 0.91 | **1.00** | -0.04 | 0.62 | **0.74** |
| **NH$_4$** | 0.43 | 0.63 | 0.43 | 0.41 | **0.78** | 0.00 | 0.24 | 0.04 | -0.04 | **1.00** | -0.02 | -0.23 |
| **Na** | 0.53 | 0.53 | 0.43 | 0.52 | 0.54 | **0.74** | 0.04 | 0.68 | 0.62 | -0.02 | **1.00** | 0.68 |
| **Cl** | 0.48 | 0.35 | 0.38 | 0.47 | 0.04 | **0.80** | 0.14 | **0.70** | **0.74** | -0.23 | 0.68 | **1.00** |

*Table 2. Concentrations of major ions (µg m$^{-3}$) were measured at the Bhola Climate Observatory-Bangladesh (BCOB), Maldives Climate Observatory-Hanimaadhoo (MCOH), Maldives Climate Observatory-Gan (MCOG) from November 2017 to March 2018.*

| Site | Na$^+$ | Cl$^-$ | NO$_3^-$ | NH$_4^+$ | nss-SO$_4^{2-}$ | nss-K$^+$ | Ca$^{2+}$ |
|---|---|---|---|---|---|---|---|
| **BCOB** | 0.4 ± 0.3 | 1.9 ± 1.8 | 7.6 ± 7.3 | 3.8 ± 3.2 | 11 ± 5 | 2.4 ±1.2 | 0.1 ± 0.1 |
| **MCOH** | 0.8 ± 0.5 | 0.7 ± 0.5 | 0.1 ± 0.0 | 4.2 ± 2.7 | 11 ± 7 | 0.4 ± 0.3 | 0.1 ± 0.0 |
| **MCOG** | 1.0 ± 0.9 | 0.8 ± 0.8 | 0.3 ± 0.4 | 0.5 ± 0.8 | 3.0 ± 2 | 0.1 ± 0.1 | 0.2 ± 0.2 |
| | | | Only during the synoptic period (18 December 2017 to 8 February 2018) | | | | |
| **BCOB** | 0.5 ± 0.4 | 3.0 ± 1.9 | 12 ± 7.7 | 2.5 ± 3.5 | 12 ± 5 | 2.9 ± 1.0 | 0.1 ± 0.1 |
| **MCOH** | 1.0 ± 0.5 | 0.6 ± 0.4 | 0.1 ± 0.0 | 4.8 ± 3.6 | 16 ± 7 | 0.5 ± 0.3 | 0.1 ± 0.0 |
| **MCOG** | 1.1 ± 1.1 | 0.6 ± 0.4 | 0.3 ± 0.3 | 0.9 ± 0.1 | 4.7 ± 4 | 0.1 ± 0.1 | 0.2 ± 0.2 |

Table S2: Why do NO3, Ca, and Mg are not correlating with any other species?
Please refer to our previous response to the comment above.

Table S3: Why does K not correlating with any other species?
Please refer to the response to the earlier comment.

**Minor Comments:**

L53-56: add appropriate references from this region.
We have added more references in the revised version.
line 64: *Ram and Sarin, 2015; Gustafsson & Ramanathan, 2016; Venkataraman et al., 2020*
line 66: *Gustafsson and Ramanathan, 2016; Dasari et al., 2019; Budhavant et al., 2015, 2023*

L120: Eq 3: How was the babs measured? Why at 365 nm only?

The absorption spectrum of water extracts was measured using a Hitachi absorption spectrophotometer-2010, in the range of 190 to 1200 nm (as described in the Methods section; lines 139-140). BrC is an organic aerosol moiety that absorbs light of shorter wavelengths. It is mainly absorbed in the ultraviolet and near-visible wavelengths, which gives it a brownish or yellowish appearance. To accurately measure the levels of BrC present in particles, the absorption coefficient between 360 and 370 nm (average of 365 nm) is commonly used and allows for inter-comparison of BrC between studies.

L255-257: There are numerous recent studies and MAC365 shall be compared with the recent studies.

In the revised version of the manuscript, we have compared our results with even earlier reports of this property. Page 10, lines 307-308.

"*The average BrC MAC$_{365}$ measured during this study was lower than values reported from close to sources in megacities such as the 1.8 m$^2$ g$^{-1}$ for Beijing winter (Cheng et al., 2011), 1.6 ± 0.5 m$^2$ g$^{-1}$ for Delhi (Kirillova et al., 2014), 1.6 ± 0.1 m$^2$ g$^{-1}$ for Kanpur (Choudhary et al., 2016), 1.5 ± 0.2 m$^2$ g$^{-1}$ for Kathmandu (Chen et al., 2020).*"

L262-264: mention the wavelength range used for AAE calculation for the better clarity for readers.

To prevent any potential interference from light-absorbing solutes such as ammonium nitrate, sodium nitrate, and nitrate ions, which have absorption peaks near 308, 298, and 302 nm, respectively, the AAE was based on the 330-400 nm range, as is common praxis. This is already mentioned in the earlier section (lines 125-126) and now in the revised manuscript (Page 5, lines 139-140).

**Author Responses and Planned Revisions to Reviewer (#2) Comments**

The Budhavant manuscript describes the results of SAPOEX campaign at three sites in South Asia. There is potential with this dataset, but the authors need to do more to connect across the results in each section (e.g. back trajectory analysis, aerosol composition, optical properties, radiative forcing). For example, the absolute concentrations of aerosol composition were only presented as campaign averages, which made it difficult to compare to the aerosol radiative forcing which was presented and discussed as an average and by month. There was often an over-simplification of the results, for example, the assertion that the MCOH site received transported air masses from the BCOB site: this did not seem to be always the case so it added confusion when discussing composition and aging. Overall the manuscript needs more refining of focus and connection among the different sections. Even in the introduction, the discussion of source and processing impacts on BC and BrC was over-simplified and lacking in precision.

We appreciate this constructive feedback and concrete suggestions to enhance our manuscript. Using these as guidance, we have thoroughly revised the manuscript and provided more information to support our findings. To address the first/major point, the revised ms is now providing also monthly average concentrations (Supplementary nformation Table S4), which are also discussed in the main ms (page 6, line 201; page 10, line 334, 337). We have further scrutinized for opportunities to relate the parts closer to each other and to synthesize the overall findings.

**Table S4.** *Monthly average concentrations of black carbon (BC), organic carbon (OC), and major ions (µg m⁻³) were measured at the Bhola Climate Observatory-Bangladesh (BCOB), Maldives Climate Observatory-Hanimaadhoo (MCOH), Maldives Climate Observatory-Gan (MCOG) from November 2017 to March 2018.*

| Site | OC | EC | Na$^+$ | Cl$^-$ | NO$_3^-$ | NH$_4^+$ | nss-SO$_4^{2-}$ | nss-K$^+$ |
|------|-----|-----|------|------|------|------|-----------|---------|
| **November-2027** | | | | | | | | |
| **BCOB** | 24.4±11 | 3.3±1.0 | 0.3±0.2 | 1.9±1.1 | 9.9±7.2 | 3.3±4.0 | 10.0±5.1 | 3.1±1.2 |
| **MCOH** | 0.4±0.1 | 0.2±0.1 | 0.5±0.2 | 0.1±0.0 | 0.1±0.1 | 3.4±0.2 | 12.5±0.7 | 0.4±0.1 |
| **MCOG** | 0.7±0.2 | 0.2±0.1 | 2.1±0.5 | 3.2±0.8 | 1.7±1.0 | 0.0±0.0 | 2.0±1.0 | 0.6±0.2 |
| **December 2017** | | | | | | | | |
| **BCOB** | 23.6±8.1 | 3.8±1.4 | 0.4±0.2 | 2.2±1.5 | 5.9±3.5 | 2.8±2.3 | 7.9±3.4 | 2.7±1.1 |
| **MCOH** | 2.8±1.8 | 0.9±0.4 | 0.9±0.4 | 0.1±0.1 | 0.1±0.0 | 3.5±1.8 | 12.9±6.0 | 0.3±0.2 |
| **MCOG** | 0.9±0.7 | 0.3±0.2 | 0.4±0.4 | 0.6±0.6 | 0.2±0.3 | 0.5±0.5 | 1.8±1.7 | 0.1±0.1 |
| **January 2018** | | | | | | | | |
| **BCOB** | 32.2±4.4 | 3.8±0.4 | 0.6±0.4 | 3.9±1.8 | 16.5±7.4 | 1.8±1.8 | 13.7±4.7 | 3.3±1.0 |
| **MCOH** | 2.7±1.5 | 1.1±0.6 | 1.1±0.6 | 0.2±0.2 | 0.1±0.0 | 5.2±4.5 | 16.1±7.7 | 0.5±0.4 |
| **MCOG** | 1.2±0.5 | 0.7±0.2 | 1.5±1.4 | 0.6±0.3 | 0.4±0.4 | 1.8±1.0 | 8.3±3.9 | 0.2±0.1 |
| **February 2018** | | | | | | | | |
| **BCOB** | 12.7±5.8 | 2.1±0.7 | 0.4±0.4 | 0.5±0.5 | 4.7±4.7 | 6.1±1.8 | 12.5±4.1 | 1.5±0.6 |
| **MCOH** | 2.4±1.1 | 1.2±0.4 | 0.8±0.4 | 0.1±0.2 | 0.1±0.0 | 4.5±1.6 | 16.7±5.7 | 0.5±0.2 |
| **MCOG** | 0.8±0.3 | 0.4±0.2 | 1.7±0.7 | 0.4±0.3 | 0.4±0.3 | 0.1±0.0 | 3.4±1.7 | 0.1±0.1 |
| **March 2018** | | | | | | | | |
| **BCOB** | 11.3±8.4 | 2.3±1.0 | 0.3±0.1 | 1.4±1.5 | 3.6±4.0 | 7.0±2.1 | 13.5±2.3 | 1.8±0.8 |
| **MCOH** | 1.4±0.9 | 0.9±0.5 | 0.5±0.4 | 0.1±0.2 | 0.1±0.0 | 3.5±2.6 | 12.2±9.6 | 0.4±0.4 |
| **MCOG** | 0.2±0.2 | 0.1±0.0 | 0.6±0.5 | 0.5±0.5 | 0.1±0.1 | 0.1±0.2 | 1.0±0.8 | 0.0±0.0 |

**Detailed comments:**

Ln 25: can this be better linked in the abstract to the aerosol optical properties?

We have the revised manuscript abstract regarding this aspect. Page 1, lines 22-23 and 27-28.

*"This likely reflects a scavenging fractionation resulting in a population of finer BC with higher $MAC_{678}$ having higher longevity."*

*"The results of this synoptic study over the large South Asian scale contribute rare observational constraints on optical properties of ambient BC (and BrC) aerosols over regional scales away from emission sources."*

Ln 46: this is an awkward phrase here. please edit

We have revised this section for clarity. Page 2, lines 48-52

*"Perhaps most concerning, anthropogenic carbonaceous aerosols have been linked to the melting of the Himalayan glaciers (Ramanathan et al., 2007; Ramachandran et al., 2023). This is especially significant because the Himalayas' watershed serves over 3 billion people, making it one of the most important water resources in the world."*

Ln 47: is this referring specifically to this region? this is undoubtedly true for BC, but WSOC may have other sources? e.g. biogenic and SOA?

Thank you for bringing this additional dimension up. We acknowledge that WSOC may have additional sources, such as biogenic SOA, and we have made the necessary changes to the manuscript. Page 2, lines 55-59

*"BrC is predominantly produced by burning fossil fuels and biomass. It can also be generated through low-temperature oxidation of biogenic substances, the polymerization of their by-products, reactions involving dienes, and the atmospheric processing of anthropogenic or biogenic volatile organic compounds (VOCs) in the presence of NOx (Andreae and Gelencser, 2006; Laskin et al., 2015)."*

Ln 47 – 56: this paragraph is difficult to follow as written. the authors need to clarify their purpose here. There is some confusion as they are trying to simultaneously discuss BC and WS-BrC. It doesn't really work and needs editing for clarity.

We recognize that our initial text was blurred. We have edited thoroughly, breaking the paragraph into two separate sections for the two aspects (Page 2, lines 53-62). The answer to the earlier comment provides more details.

Ln 49: This sentence is awkward and the logical transition here is unclear

We have addressed the issue. Page 2, lines 60-63

*"As per the current understanding, BC displays comparatively low reactivity and undergoes negligible changes over long distances. On the other hand, BrC seems to be subject to bleaching (Dasari et al., 2019). It is, therefore, imperative to delve into the dynamics of the optical properties of BrC during its long-distance transport."*

Ln 65: edit for clarity

We have edited for clarity. Page 3, lines 77-79.

*"The South Asian Pollution Experiment 2018 (SAPOEX-18) was a large international campaign to study BC and BrC absorption properties during long-range transport in the South Asian source-receptor system using multiple approaches and sites."*

Ln 88: What is the particle size here? TSP? PM2.5

We have used $PM_{2.5}$ samples. This is now explicitly mentioned here. Page 4, line 100

Ln 111: are there any concerns about the high loading on these filters? typically filter-based photometers limit the filter loading to that which corresponds to a 50% transmission. the filters collected for this offline analysis would not have their loading limited by light transmission. i understand that this correction is intended to address the filter loading, but these correction schemes were originally designed for online instruments which have a filter advancement/change at a set transmission threshold. can this concern be addressed?

Yes, the filter loading is a concern in this highly polluted air (especially for BCOB). We have invested substantial consideration into any effects of this and have thus previously developed and discussed filter-loading corrections and these are cited in this manuscript (e.g., Budhavant et al., 2020). Page 4, lines 125-129

Ln 180-182: its not clear how these facts are relevant here. please remove or expand the discussion to make this more clear.

The revised manuscript has had these irrelevant parts removed.

Ln 191-195: what about biogenic SO4? do you have a constraint on the possible marine contribution that goes beyond sea salt? Additionally, this rationale of sources from central and east India is a bit confusing as the BTs indicate that air masses predominantly leave the Indian subcontinent near BCOB or from west India before traveling to MCOH. If the aerosol composition from the west side of India is markedly different (e.g. higher SO4 fraction) than the IGP and BCOB, than the aging discussion of MCOH representing aged BCOB aerosol needs to be more refined.

We have included additional information on biogenic sulfate in the updated manuscript, page 7, lines 223-224. We have evaluated the SO4/BC load for samples coming from West India vs. the Bay of Bengal, page 8, 255-264. The "synoptic" comparison focuses more on winds from the IGP and southern India. The revised manuscript is updated to this effect. Also, we have already addressed this in our previous comments.

*"Furthermore, a previous study at MCOH found that dimethyl sulfide (DMS) contributes only up to 3% to nss-$SO_4^{2-}$ in polluted air (Granat et al., 2010)."*

*"Our observations have shown an increase in the $SO_4^{2-}$/BC ratio when aerosols are transported from IGP. Notably, this ratio is more pronounced at MCOH than at BCOB (Figure 5). This shift in composition likely signifies the generation of secondary sulfate from anthropogenic $SO_2$ during extended transportation. It was observed that there was a lower $SO_4^{2-}$/BC at MCOG than at MCOH. This might be because $SO_4^{2-}$ gets washed out more easily than BC in this region (Budhavant et al., 2020), and another factor is that MCOG has slightly different AMBT paths than MCOH (Figure 1). It is worth noting that there could be some minor sources of emissions along the route to the receptor sites, like ships and small islands. However, their impact on the overall regional loading of BC is insignificant. Therefore, we can infer that most BC loading over the northern Indian Ocean originates from high-emission source areas in South Asia. While there was an increase in the $SO_4^{2-}$/BC ratio, two other important coating components, WSOC/BC and WIOC/BC, declined."*

Ln 208: Is the EC supposed to be BC?

We confirm that this is BC—revised manuscript line 249.

Ln 217-223: I'm still stuck on the SO4 discussion. the provenance of the SO4 seems very relevant in determining if these 3 sites do represent different ages of the same air mass. as briefly mentioned in this section, increased SO4 would also seem to be very relevant for the coatings question. however some of the previous discussion of the loss of water soluble fraction during aging and transport (WSOC) seems to conflict with this. certainly the increase in SO4 at MCOH, absolute concentration as well as an extreme increase in relative contribution, is very relevant for coating of BC and internal mixture of BC and SO4 aerosol. i would like to ee more discussion of this potentially conflicting observations between wsoc and so4, and using the three sites as steps in the aging process of one air mass. If the rationale for higher SO4 at MCOH is a shift in geographic source region, than the discussion of aging of two different aerosol systems needs to be included.

We have already addressed this in our previous comment, pages 8, lines 255-264, and 9, 272-283.

We have already addressed this in our previous comment.

Ln 265-267: are these differences in AAE signficant? the std deviation is relatively high.

We have noted the differences mentioned, but they do not seem significant. Hence, we have revised this sentence in the revised manuscript. Page 10, line 316-317.

Ln 282-284: can the authors discuss why the atmospheric forcing was higher at MCOH while the surface concentrations for summed species was lower? it is also a bit difficult to interpret the relationship between the aerosol radiative forcing and the rest of the surface aerosol discussion as the time scales are not well aligned for aerosol concentrations and the ARF. overall, i'd like to see better connections among the sections of the discussion.

In general, MCOH has lower radiative forcing than BCOB. However, MCOH had slightly higher atmospheric forcing during January due to outflow from IGP, as shown in Figures 5, 2, and 3. We have provided more clarity in the revised version.

Page 10-11, lines 337-344 and 346-350.

*"During winter, the IGP experiences a significant increase in aerosol loading, mainly carbon aerosols resulting from fossil fuel and biofuel combustion (Gustafsson et al., 2009; Kaskaoutis et al., 2014; Dasari et al., 2020). As spring progresses, dust becomes the dominant aerosol in the northwestern region of India and arid areas of southwest Asia (Kaskaoutis et al., 2014; Singh et al., 2014; Dumka et al., 2023). At the same time, significant agricultural burning in Southeast Asia results in significantly elevated concentrations of carbonaceous aerosols (Kaskaoutis et al., 2014; Budhavant et al., 2015; Bikina et al., 2019). Also, Himalayan forest fires and wheat residue burning in the IGP contribute to the aerosol burden during spring (Gautam et al., 2007; Bikina et al., 2019)."*

*"The findings are consistent with our earlier study conducted at MCOH, which showed that anthropogenic aerosols, such as BC, BC, $nss-K^+$, $nss-SO_4^{2-}$, and $NH_4^+$, were predominantly in the fine mode (70-95%) and particularly observed in the air masses coming from IGP during the period (Budhavant et al., 2018). Therefore, it is crucial to address this issue and take appropriate measures to reduce the amount of anthropogenic aerosol loading."*

Table 1: this AAE is calculated off a very narrow range in wavelength. is 400 nm the longest avelength measured here? What are the potential shortcomings of reporting AAE for such a narrow range in wavelength? Also, it is difficult to assess the trends in the ambient concentrations when only the averages for the entire period are reported. It is fine to present

the ratios in the figures, but useful to also be able to see changes in the absolute concentrations as well.

We have measured the wavelength range from 190 to 1200 nm, yet, as elaborated above in response to reviewer 1, the AAE for WS-BrC is customarily reported for the range of 330 to 400 nm. One reason is to prevent any potential interference from light-absorbing solutes such as ammonium nitrate, sodium nitrate, and nitrate ions, which have absorption peaks near 308, 298, and 302 nm. Further, AAE is more dependent on linear ratios for shorter wavelengths, while the correlation is weaker for longer wavelengths. We have revised and included these motivations in the additional information for Table 2 and Table S4.